# Anisotropic surface potentials induced by competitive ion adsorption enable the synthesis of branched cubic Pt mesocrystals

Yuna Bae [1], Eun Mi Kim [2], Jaehun Chun [1], Zihua Zhu [3], Trevor H. Moser[3], Hanlei Zhang[1], Jaeyoung Heo[1], Yun Kyung Shin [4], Hua Zhou [5], James E. Evans [3], Emil C. S. Jensen[6], Kristian S. Mølhave [7], Kristen A. Fichthorn [2] ✉, James J. De Yoreo [1,8] ✉ & Dongsheng Li [1] ✉

Creation of complex nanostructured materials through oriented attachment (OA) requires the manipulation of interparticle forces, including electrostatic repulsion, which depends strongly on surface potentials and can be modified through the effect of solution environment on interfacial chemistry. Here we show that time-dependent anisotropies in surface potential driven by competitive ion adsorption can alter facet-selectivity during OA. This phenomenon enables the synthesis of branched cubic Pt mesocrystals. Initially, Pt nanoparticles attach preferentially at their {100} facets to form a well-defined cubic core. Over time, changes in ion adsorption shift the attachment preference to the {111} facets, promoting branch formation. In both stages, anisotropic surface potentials generate electrostatic torques that align the particles prior to attachment. These findings demonstrate a generalizable strategy for directing the architecture of nanomaterials through time-resolved control of interfacial chemistry during OA, offering new pathways for the design of complex mesoscale structures.

When crystalline materials form through assembly of nanoparticles (NPs), their properties are strongly influenced by the assembled architecture because phenomena like photon and electron scattering, electron–hole recombination, and dislocation generation depend on the material's characteristic length scales and topology[1-3]. Oriented attachment (OA) has emerged as a key pathway for creating single-crystal-like structures with diverse morphologies[4-7].

During OA, neighboring particles align and fuse along matching crystallographic planes, but intriguingly, they often do so with high facet-selectivity—certain crystallographic facets are repeatedly favored over others[4,8,9]. This consistent selectivity is surprising because

particles in solution constantly undergo random Brownian motion, leading to sampling of all possible orientations. Why particles consistently select specific crystallographic facets when attachment on any set of matched lattice planes would reduce the systems energy remains an unresolved question in the field. Answering that question would address one of the key challenges in designing crystal structures: understanding how to control and select specific facets for OA, thus enabling the tailoring of OA-based crystal growth and assembly.

Recent efforts have begun to unravel the complex interplay of interparticle forces that drive OA. For example, real-time imaging of OA in the ZnO system has established a relationship between particle

[1]Physical Sciences Division, Pacific Northwest National Laboratory, Richland, WA, USA. [2]Department of Chemical Engineering, The Pennsylvania State University, University Park, PA, USA. [3]Environmental Molecular Sciences Laboratory, Pacific Northwest National Laboratory, Richland, WA, USA. [4]Department of Mechanical Engineering, The Pennsylvania State University, University Park, PA, USA. [5]Advanced Photon Source, Argonne National Laboratory, Lemont, IL, USA. [6]Insight Chips ApS., Kgs. Lyngby, Denmark. [7]Center for Nanofabrication and Characterization, Technical University of Denmark, Kgs. Lyngby, Denmark. [8]Department of Materials Science and Engineering, University of Washington, Seattle, WA, USA. ✉e-mail: fichthorn@psu.edu; James.DeYoreo@pnnl.gov; Dongsheng.Li2@pnnl.gov

structure, interaction forces arising from ion-solvent correlations and dipolar interactions, and the resulting assembly dynamics[10]. Other studies in metal oxide systems have explored the influence of ion-correlation forces on OA[11]. These studies show that OA kinetics and pathways are intricately tied to interfacial chemistry, which is governed by environmental factors, including electrolyte type and concentration, surface adsorbates, and pH[12–17]. However, despite this progress, current understanding remains largely focused on interaction energetics, with little attention paid to how directional or rotational alignment is achieved. Since facet-selective OA requires precise orientation at the moment of collision, this gap points to the need for a mechanism capable of generating directional torque, especially at nanometer distances.

Amongst the forces defining interparticle potentials, the repulsive electrostatic force is most strongly dependent on the chemistry of the NP surface as it relies on the surface potential, which can vary independently on different crystal facets[18,19]. Here we show that the facet-specificity of attachment by Pt NPs can be manipulated through changes in the surface potential of distinct facets to drive a transition from attachment on Pt{100} to Pt{111}. This transition leads to a switch from growth of a cubic core through {100} attachment to the extension of {111}-oriented branches out from the cubic core through {111} attachment. Moreover, we find that the disparity in the surface potential between the two facets creates an electrostatic torque that is critical for ensuring facet specificity. Finally, we demonstrate that competitive ion adsorption underlies the changes in surface potential that lead to the transition from {100} to {111} attachment through torque-driven facet selection.

## Results

### Synthesis and structural analysis of branched Pt cubes

Branched Pt cubes (50–120 nm, Fig. 1a, Supplementary Fig. 1) were obtained by mixing potassium tetrachloroplatinate (K$_2$PtCl$_4$) and formic acid (HCOOH) in deionized water at room temperature for ≈1 h. The Pt NPs form by ≈10 min after mixing (Supplementary Fig. 2) and transform into branched cubes after 1 h. The diffraction streaks in selected area electron diffraction (SAED) from a single branched cube confirm the constituent NPs are attached along the same crystallographic orientation with slight misalignments (Fig. 1b). Many nanorods protrude from the cube's faces, mostly aligned along its diagonals (Fig. 1a, c, Supplementary Fig. 1).

Collection of samples at earlier time points (≈45 min) reveals mesocrystalline cubes without branches, several tens of nm in size and composed of ≈3 nm NPs (Fig. 1d). High-resolution (HR-) transmission electron microscopy (TEM) images and fast Fourier transform (FFT) patterns show that the cubes are nearly single crystals composed of NPs attached on {100} planes, indicating they form by {100} OA of primary NPs (Fig. 1d, e). Additionally, we observe edge dislocations (an extra half {100} plane, Supplementary Fig. 3), frequently seen at the attaching interfaces during imperfect OA events[7,20], further indicating that the cubes form via {100} OA. By 1 h, 3 nm diameter branches composed of single NPs attached along {111} are formed (Fig. 1f, g). Thus, we find that these branched cubic Pt mesocrystals form via OA of ≈3 nm primary NPs, starting with {100} OA to create cubic cores, followed by {111} OA to form the branches (Fig. 1h). Here, the mesocrystals are composed of the primary Pt NPs, which are crystallographically aligned and individually identifiable, but are not spatially separated as a consequence of OA.

### Early-stage growth and structural evolution

Nanochannel liquid-phase TEM (LPTEM) performed during the early-stage of growth (Supplementary Movies 1–3)[21,22], shows that primary NPs nucleate, grow to a uniform size of ≈3.1 nm, and then aggregate into small clusters (Fig. 2a, Supplementary Fig. 4, and Supplementary Movie 1). The aggregated NPs are separated by a solvent layer (1.7 ± 0.5 nm)[23] but remain close to each other despite their Brownian

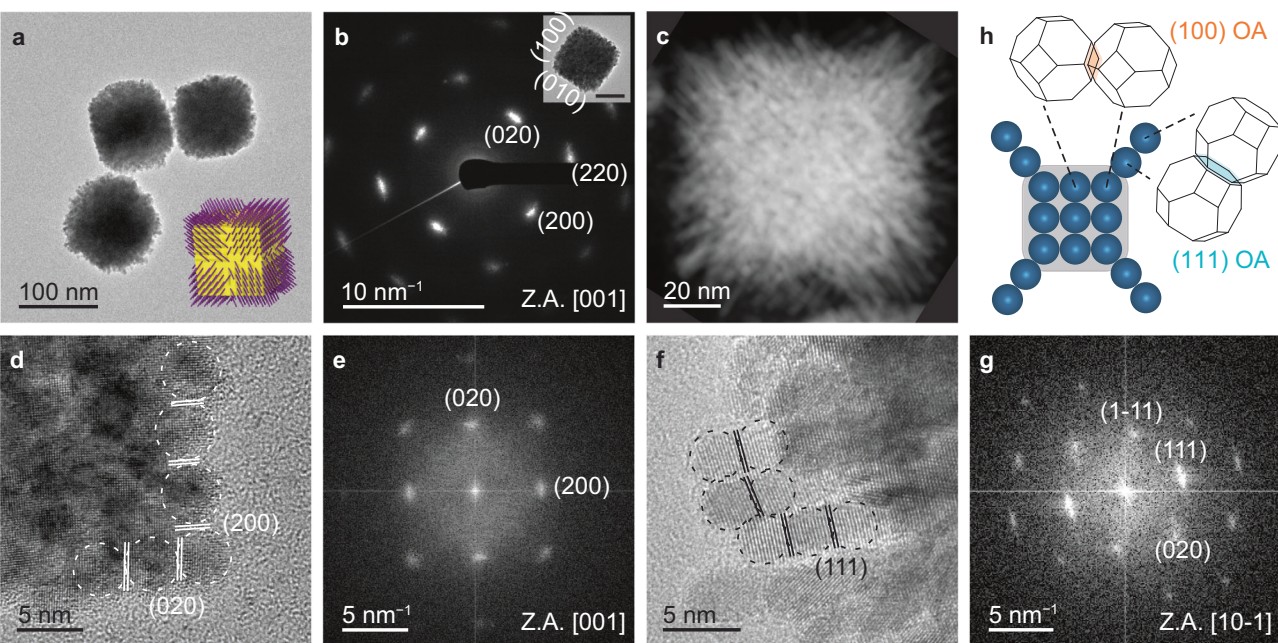

**Fig. 1 | Morphological analysis of branched cubic Pt mesocrystals. a** Ex situ TEM image of synthesized mesocrystals formed from 1.45 mM of K$_2$PtCl$_4$ and 52 mM of HCOOH. Inset, schematic of branched cubic mesocrystals showing the cube-shaped core (yellow) and branches (purple) growing from the cube faces. **b, c** SAED pattern (**b**) and Scanning TEM image (**c**) from a single mesocrystal. Inset in (**b**), corresponding TEM image. Scale bar for the inset: 40 nm. **d, e** HR-TEM images (**d**) and corresponding FFT patterns (**e**) of cubic core obtained at 45 min, with an outline showing {100} attachments between NPs. **f, g** HR-TEM images (**f**) and corresponding FFT patterns (**g**) of branches showing they result from {111} attachments between NPs to form {111}-aligned nanorods. Branched cubes were obtained at > 60 min NPs are indicated by dotted circles, and the attachment plane is indicated by solid lines in (**d, f**). Images in (**a–c**) and (**g**) are observed 2 days after the reaction. Z.A., zone axis. **h** Schematics showing the OA direction and the resulting morphology. Pt NPs are depicted as dark blue spheres, with the attaching {111} and {100} facets colored blue and orange, respectively.

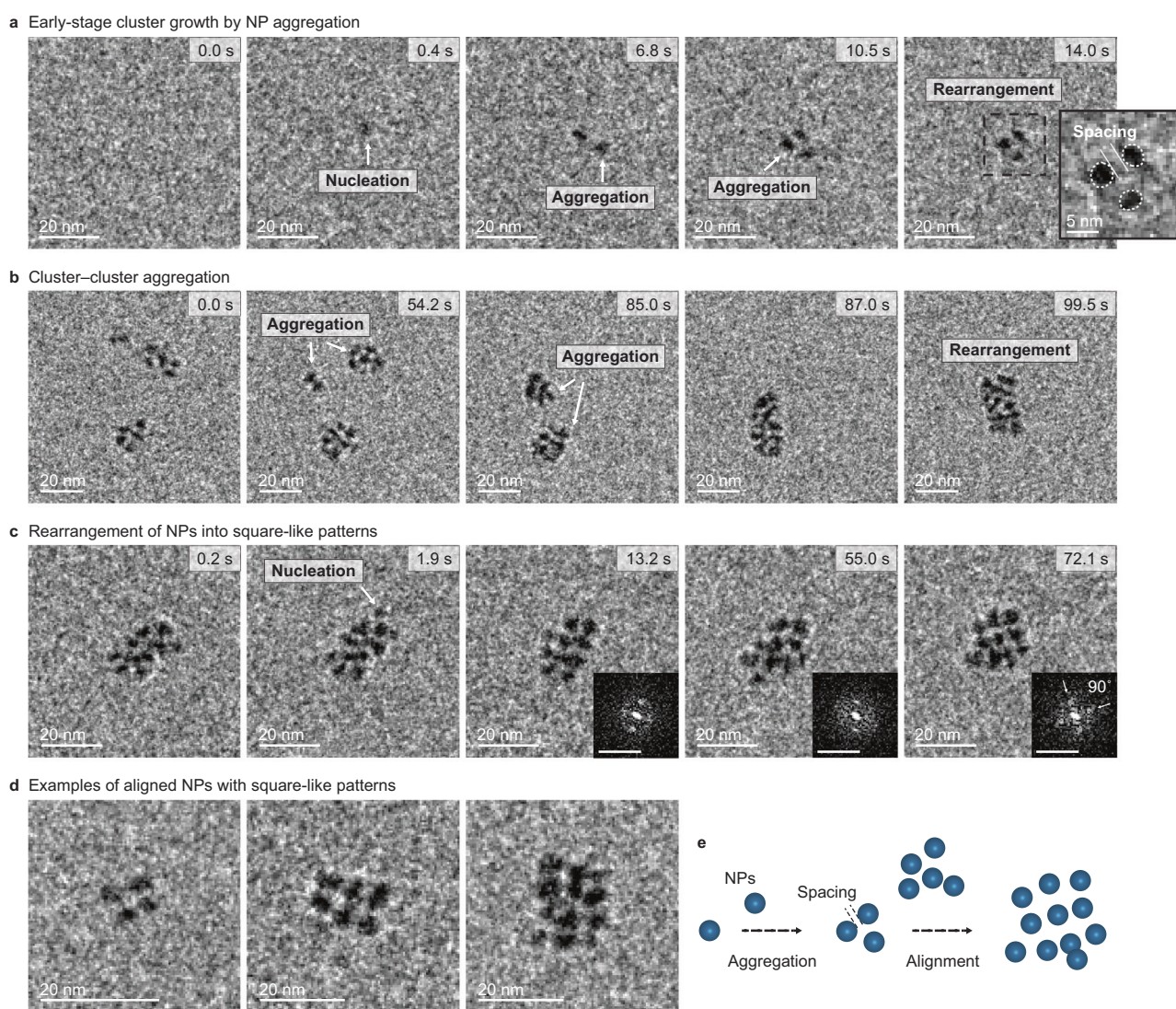

**Fig. 2 | Cluster formation by NP aggregation and pre-alignment of solvent-separated NPs. a–c**, Sequential LPTEM images showing early-stage cluster growth by NP aggregation (**a**), further growth by cluster aggregation (**b**), and rearrangement of NPs into square-like patterns (**c**). Insets, corresponding FFT patterns of each TEM image in **c**. Scale bars of insets FFT patterns: 0.5 nm⁻¹. **d** Representative LPTEM images of aligned NPs into square-like patterns within clusters. **e** Schematic of the early-stages growth process showing NPs aggregating and aligning.

motion, creating a 2D pattern, because they are attracted to–but not adhered to–the nanochannel surface. The clusters further grow by aggregation with other clusters or addition of individual NPs (Fig. 2b, Supplementary Movie 2). The NPs are initially randomly aligned but gradually reorient into an ordered square pattern (Fig. 2c, Supplementary Movie 3), which may reflect the initial step in forming the cubic core via {100} OA (Fig. 2d). Though the crystallographic orientations cannot be resolved in the LPTEM images, SAED patterns of large clusters show either random orientation or internal ordering (Supplementary Fig. 5). These LPTEM results confirm that the NPs form rapidly and serve as building blocks in early-stage growth, first associating and then rearranging before OA begins (Fig. 2e). While the LPTEM captured the early stage of growth, the system did not evolve into the final branched cubes. This limitation is likely due to the confined liquid-cell environment, either because the volume is too small for an adequate number of NPs to be generated for the particles to reach the branching stage and/or because restricted Brownian motion –caused by interactions with the SiN_x window– and increased viscosity in the ≈100–200 nm-thick liquid layer suppressed the interparticle dynamics required for further growth, particularly near the periphery of ≈100 nm-sized NP clusters[24].

Cryogenic (cryo-) TEM confirms the initial stages seen in LPTEM in the absence of the electron beam and provides the time-dependent evolution of the branched cubes (Fig. 3). In the earliest stage (stage i), clusters of tens of nm, composed of ≈3 nm solvent-separated NPs, are captured (Fig. 3a, b, Supplementary Fig. 6), consistent with LPTEM (Fig. 2, Supplementary Figs. 4, 5, 7). SAED patterns of a cluster taken at 15 min show faint diffraction rings, indicating NPs aggregation with random orientations (Fig. 3f). In some regions, there are NPs arranged in a square pattern (Supplementary Fig. 8) and NPs aligned along ⟨100⟩ directions, as well as other orientations (Fig. 3k–n). These observations imply that NPs within a cluster gradually organize out of a disordered arrangement into a square pattern with their {100} facets aligned.

At ≈20 min (stage ii), a dense core begins to form, surrounded by a shell of loosely associated NPs (Fig. 3c). Based on the HR-cryo-TEM image, FFT diffraction spots for the core and diffraction rings for the shell confirm that the core consists of orientationally aligned/attached NPs while the shell contains randomly aligned NPs (Fig. 3o–q). This indicates that OA of NPs begins within the interior of the cluster, forming a high-contrast mesocrystalline core. The dense core grows into a cubic shape via continued OA of NPs from the shell (Fig. 3d).

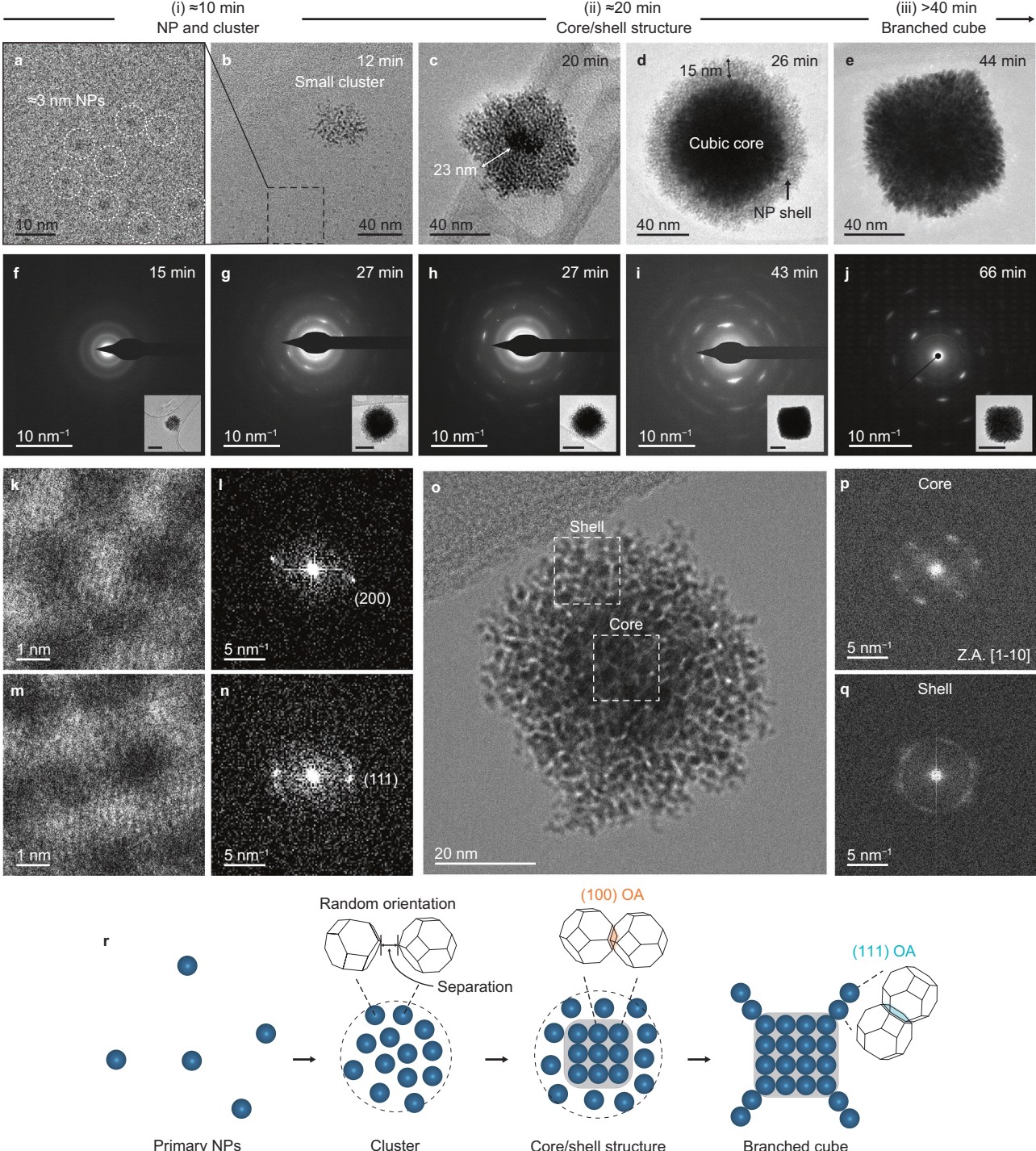

**Fig. 3 | Evolution of branched cubic mesocrystals via OA of primary NPs.**
**a–e** Representative cryo-TEM images at different reaction times showing the multi-step crystallization of Pt mesocrystals going from NPs to clusters (stage i), to core/shell structures (stage ii), to branched cubes (stage iii). **f–j** SAED patterns at the reaction time points; cluster at 15 min (**f**), core/shell structures with thick shell (**g**) and thin shell (**h**) at 27 min, cubes without branches at 43 min (**i**), and branched cubes at 66 min (**j**), respectively. Inset, corresponding cryo-TEM images of each single structure. Scale bars for the insets: 50 nm. **k–n** HR-cryo-TEM images observed at 13 min and corresponding FFT patterns showing ⟨100⟩ alignment (**k**, **l**) and ⟨111⟩ alignment (**m**, **n**) between NPs within a cluster. **o–q** HR-cryo-TEM image of a core/shell structure observed at 25 min (**o**) and corresponding FFT patterns of the core (**p**) and shell (**q**) marked by dashed lines in (**o**). Z.A., zone axis. **r** Schematic showing the evolution of branched cubic mesocrystals via OA involving a switch in the facet upon which attachment occurs.

SAED patterns of the core/shell structures show that single-crystal-like diffraction streaks appear atop the diffraction rings, with rings vanishing as the shell diminishes (Fig. 3g–i). After 40 min (stage iii), nanorods begin to grow outward on the faces of the cubic core, resulting in a branched cube with a single-crystal-like diffraction pattern (Fig. 3e–j). The SAED patterns of intermediate structures from early to final growth stages demonstrate an increase in internal order from clusters to core/shell structures to branched cubes, further supporting the growth model through gradual OA of loosely associated NPs (Supplementary Figs. 5, 9, 10).

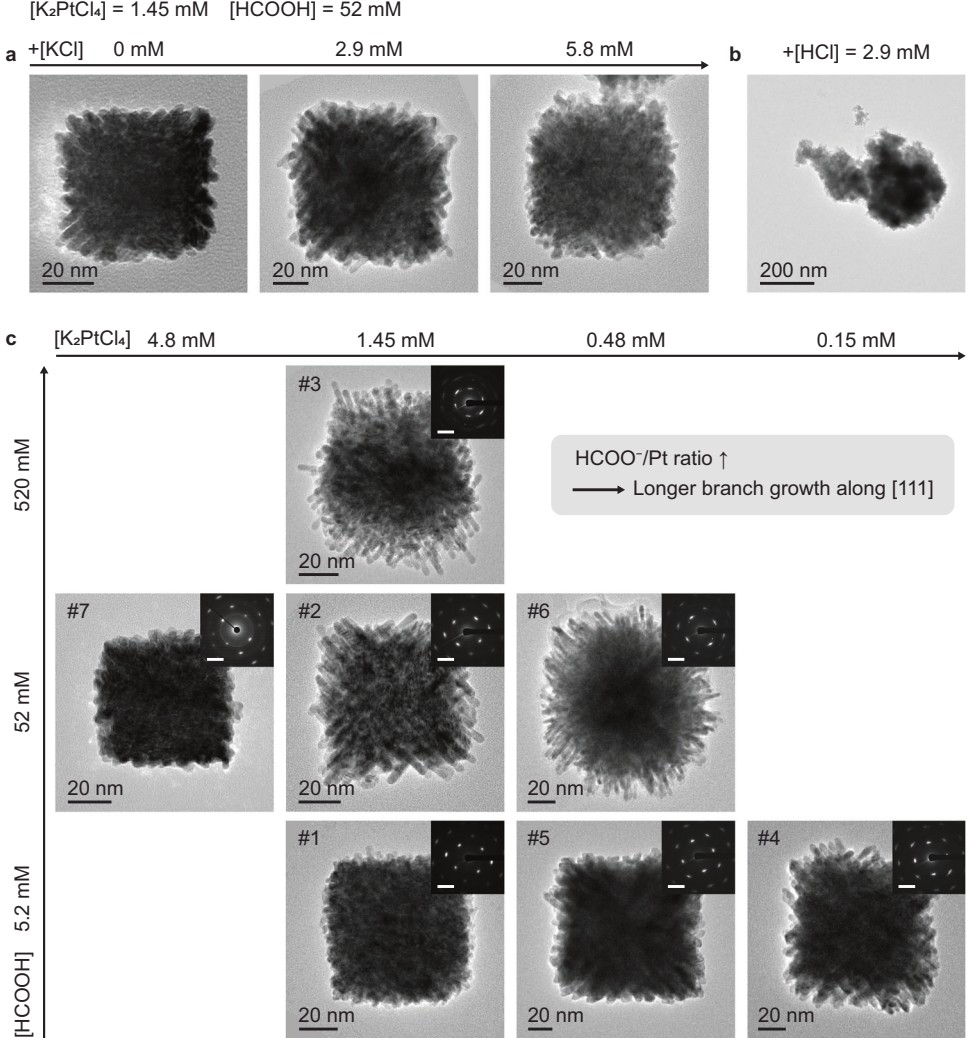

**Fig. 4 | Effect of ionic additives and precursor concentrations on the shape of branched cubic Pt mesocrystals. a, b** TEM images showing the effect of adding KCl or HCl on the resulting shape. The concentration of KCl added is 0, 2.9, and 5.8 mM, respectively in (**a**), and the concentration of HCl added is 2.9 mM in (**b**). The concentrations of $K_2PtCl_4$ and HCOOH are 1.45 mM and 52 mM, respectively. **c** TEM images of branched cubic Pt mesocrystals as a function of initial precursor concentrations showing the formation of longer branches with increasing $HCOO^-$/Pt ratio. The synthesis conditions for each numbered image are listed in Supplementary Table 1. Insets show the corresponding SAED patterns of each TEM image. Scale bars, 5 $nm^{-1}$. All TEM images are observed 2 days after the reaction.

## The role of surface-potential in controlling NP alignment

To understand the factors controlling morphology, we investigated the effect of solution-phase species by varying $[K_2PtCl_4]$ and [HCOOH] or by adding KCl or HCl (Fig. 4, Supplementary Fig. 11 and Supplementary Table 1). Adding KCl (2.9 mM and 5.8 mM) does not disrupt the formation of branched cubes, while adding HCl (2.9 mM) leads to random aggregation (Fig. 4a, b). Branch lengths varied with the initial $[HCOOH]/[K_2PtCl_4]$ ratio, with longer branches seen at higher [HCOOH] or lower $[K_2PtCl_4]$ (Fig. 4c). Cryo-TEM shows that branch growth starts earlier at higher $[HCOOH]/[K_2PtCl_4]$ ratio (Supplementary Fig. 12), suggesting the transition from {100} to {111} attachment correlates with this ratio. Given the impact of solution composition, we hypothesize that the switch of attaching facets relates to ion adsorption on NP surfaces and the resulting changes in surface potential.

According to the following chemical reaction, the reduction of $K_2PtCl_4$ by HCOOH releases chloride ions $(Cl^-)$ and hydrogen ions $(H^+)$: $K_2PtCl_4 + HCOOH \rightarrow Pt + 2 K^+ + 2 H^+ + 4 Cl^- + CO_2$[25]. The measured pH is initially 2.50, dropping sharply at ≈10 min (stage I) and stabilizing at 2.36 ± 0.01 after 15 min (stage II–III) (Fig. 5a). Thus, within the first ≈10 min, the reaction is nearly complete and most primary NPs have been generated, as deduced from TEM observations (Supplementary

Figs. 2, 3). Consistent with this observation, the UV-Vis absorbance spectra show a sharp increase of surface plasmon resonance of Pt NPs at a wavelength of ≈200 nm, demonstrating a rapid increase in NP concentration within ≈10 min, followed by a gradual decrease after ≈24 min due to NP aggregation (Fig. 5b, Supplementary Fig. 13). After Pt reduction, the concentrations of all related ions stabilize at $[H^+] = 4.75$, $[K^+] = 2.9$, $[Cl^-] = 5.8$, and $[HCOO^-] = 1.85$ mM (Supplementary Table 1).

This indicates that the subsequent switch in attachment behavior, which occurs much later (at ≈45 min), is not caused by ongoing changes in bulk solution chemistry. Instead, it must be driven by the evolution of the NP surface chemistry itself. As the ≈3 nm NPs (Fig. 3a) attach and fuse into larger crystals (≈120 nm, Fig. 3e), the total NP surface area in solution $(A_{tot})$ decreases by ≈30 times. This drastic decrease in $A_{tot}$ available for ion adsorption would alter the surface coverage of the various ions on the different facets over time, thereby modifying their surface potential.

To test this, we measured the zeta potential $(\zeta)$—a proxy for surface potential—of surfaces dominated by {100} and {111} facets as a function of $A_{tot}$ under conditions mimicking the stage ii–iii (Supplementary Figs. 14, 15). The concentration of the initially generated NPs

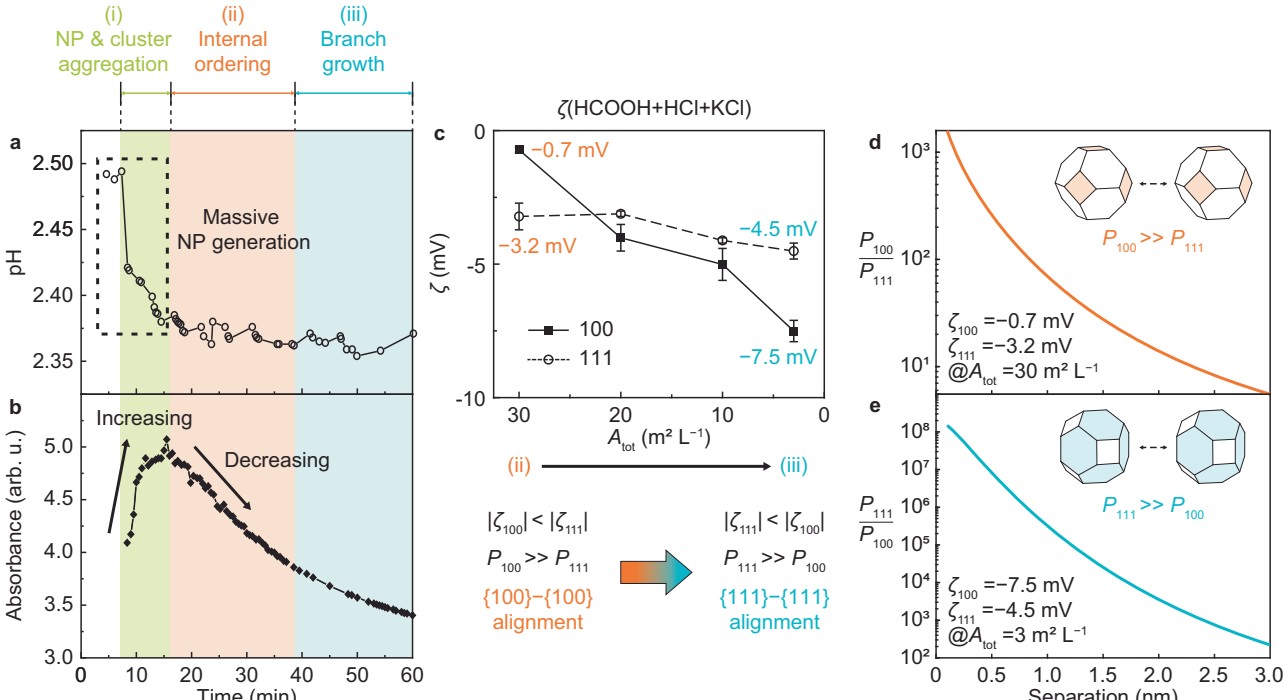

**Fig. 5 | Evolution of anisotropic surface potential due to repeated OA and transition of pre-aligning NP surface. a, b** pH (**a**) and UV-vis absorbance at 236 nm (**b**) during the growth of Pt mesocrystals at initial concentrations of 1.45 mM $K_2PtCl_4$ and 52 mM HCOOH. Reaction stages are shaded by color: light green for stage i, orange for stage ii; and blue for stage iii. **c** Zeta potential ($\zeta$) of Pt{100} versus Pt{111} as a function of $A_{tot}$ in a solution containing 2.9 mM HCl, 2.9 mM KCl, and 26.5 mM HCOOH. [H⁺], [K⁺], [Cl⁻], and [HCOO⁻] are 4.0 mM, 2.9 mM, 5.8 mM, and 1.14 mM, respectively. Data points represent the mean value from three independent measurements, with error bars indicating standard deviation. **d, e** Alignment ratio as a function of separation distance between two NPs. The ratio of probabilities for the {100} − {100} and {111} − {111} alignments ($P_{100}/P_{111}$) when $|\zeta_{100}| < |\zeta_{111}|$ (at $A_{tot} = 30$ m² L⁻¹ in (**c, d**) and the ratio of the probabilities for the {111} − {111} and {100} − {100} alignments ($P_{111}/P_{100}$) when $|\zeta_{100}| > |\zeta_{111}|$ (at $A_{tot} = 3$ m² L⁻¹ in (**c, e**). $A_{tot}$ of 30 m² L⁻¹ and 3 m² L⁻¹ represent stage ii (orange) and stage iii (blue), respectively. Source data are provided as a Source Data file.

and $A_{tot}$ are estimated to be 2.5 µM and 30 m² L⁻¹, respectively, considering the primary 3 nm NPs are truncated octahedra with {100} and {111} facets (Supplementary Fig. 16)[26]. The concentrations of H⁺, K⁺, Cl⁻, and HCOO⁻ are reproduced by mixing KCl, HCl, and HCOOH (#2 in Supplementary Table 1).

The results showing that both {100} and {111} surfaces become more negatively charged with decreasing NP concentration (Supplementary Figs. 15, 17) support the hypothesis that the surfaces are undersaturated with respect to ion adsorption, making them highly sensitive to the change in $A_{tot}$[27,28]. This sensitivity leads to a striking anisotropic behavior between {100} and {111} surfaces (Fig. 5c). At a high $A_{tot}$ (30 m² L⁻¹, representing stage ii), the {100} surface exhibits a smaller $\zeta$ value (−0.7 mV) than the {111} surface (−3.2 mV). As $A_{tot}$ decreases to 3 m² L⁻¹ (representing stage iii), $\zeta_{111}$ changes slightly from −3.2 mV to −4.5 mV, while $\zeta_{100}$ becomes more negative, dropping from near-neutral to −7.5 mV, which is then below $\zeta_{111}$. This demonstrates a clear crossover in the relative electrostatic repulsion of the two surfaces during Pt mesocrystal growth via repeated OA events.

However, the magnitude of the electrostatic repulsion alone cannot account for the observed switch in the plane of attachment for the following reasons. Both facets are negative throughout the synthesis process; consequently, the electrostatic component of the interaction potential is always repulsive. More importantly, although the relative strength switches, $\zeta_{111}$ is nearly constant. Thus, if {111} attachment is possible during late-stage growth, then it is possible at early times as well. This suggests that the anisotropy in the electrostatic interactions leads to a torque that pre-aligns the particles prior to contact.

Although such torques have been observed in other systems as a consequence of both dipole−dipole interactions[10,29] and anisotropic

van der Waals (vdW) interactions[30,31], we note that these sources of torque are unlikely to be the cause in our system (Supplementary Note 2). First, individual Pt particles are nearly spherical (truncated octahedrons, Supplementary Fig. 16) and possess a symmetric face-centered cubic (FCC) structure. Given this high degree of symmetry, significant torques originating from intrinsic material properties (e.g., dipole moment, anisotropic dielectric response) are not expected. Furthermore, any such intrinsic torques would remain constant during the growth process and therefore cannot account for the dynamic switching of the attaching surfaces.

To determine whether these anisotropic electrostatic interactions can guide OA on specific facets of NPs, we calculated the ratio of the probabilities for {100}−{100} and {111}−{111} alignments using the orientational probability distribution function described by the Smoluchowski equation, which embodies the balance between Brownian and electrostatic torques (Supplementary Note 1)[32]. The preferential alignment along surfaces with lower relative surface potential increases significantly as the NP separation decreases (Fig. 5d, e). These calculations predict that spatially separated, randomly oriented NPs within a cluster (Fig. 3f, Supplementary Fig. 8) will become aligned along {100} before attachment through rotations relative to one another that are driven by these facet-dependent electrostatic interactions and resultant torques (Fig. 5d). Similarly, when the relative values of the surface potential are later inverted, the calculations predict that electrostatic torque drives alignment of NPs along {111} surface just before attachment (Fig. 5e).

## Surface potential modulation by competitive ion competition

To understand the crossover in $\zeta$ between the {100} and {111} surfaces (Fig. 5c), we investigated the competitive ion adsorption of the key

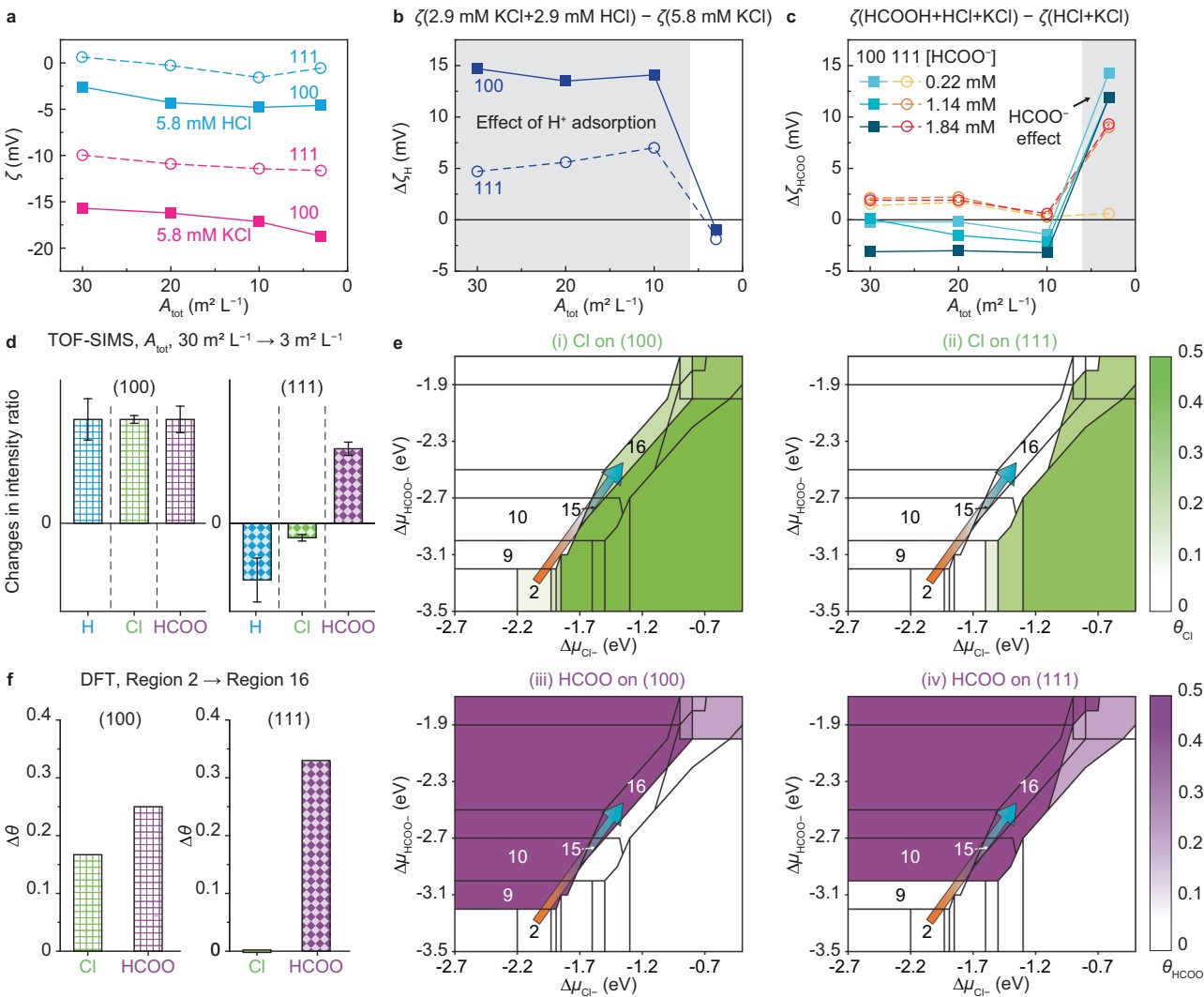

**Fig. 6 | Facet-dependent adsorption characteristics of competing ions. a** Zeta potential ($\zeta$) of Pt{100} versus Pt{111} as a function of $A_{tot}$ in 5.8 mM HCl (blue) and 5.8 mM KCl (pink) solutions. **b** $\triangle\zeta_H$, the difference between $\zeta$ in a solution containing 2.9 mM HCl and 2.9 mM KCl relative to $\zeta$ in 5.8 mM KCl solution (navy). The gray shaded area indicates the region where the effect of H adsorption is prominent. **c** $\triangle\zeta_{HCOO}$, the difference between $\zeta$ in solutions containing 2.9 mM HCl, 2.9 mM KCl, and varying [HCOO⁻] relative to $\zeta$ in a solution containing 2.9 mM HCl and 2.9 mM KCl. Data points for different [HCOO⁻] are shown in yellow ({111}) and light blue ({100}) for 0.22 mM; orange ({111}) and medium blue ({100}) for 1.14 mM; and red ({111}) and dark blue ({100}) for 1.84 mM. The gray shaded area indicates the region where the effect of HCOO⁻ addition is prominent. Squares represent the {100} facets and circles represent the {111} facets. **d** Difference in relative intensity ratio of adsorbed ions (H, Cl, and HCOO) on Pt{100} and Pt{111} by subtracting the ratio at $A_{tot}$ = 30 m² L⁻¹ from the one at $A_{tot}$ = 3 m² L⁻¹, obtained from TOF-SIMS spectra at $A_{tot}$ of 30 m² L⁻¹ and 3 m² L⁻¹ in a solution containing 2.9 mM HCl, 2.9 mM

KCl, and 26.5 mM HCOOH. [H⁺], [K⁺], [Cl⁻], and [HCOO⁻] are 4.0 mM, 2.9 mM, 5.8 mM, and 1.14 mM, respectively. Blue, H; Green, Cl; and Purple, HCOO. Data points represent the mean of four independent measurements for the {100} surface at $A_{tot}$ of 30 m² L⁻¹ and the mean of five independent measurements for each of the other conditions, with error bars indicating standard deviation. The measured adsorption levels of different ions cannot be compared. **e** Minimum surface-energy diagram as a function of the chemical potentials of Cl⁻ and HCOO⁻ relative to the gas phase ($\triangle\mu_{Cl⁻}$ and $\triangle\mu_{HCOO⁻}$) on Pt(100) and Pt(111), as calculated by DFT. $\theta$ is the coverage of each species. The colored regions indicate the different $\theta$ shown in the color bar on the right: green for $\theta_{Cl}$ and purple for $\theta_{HCOO}$. (i) $\theta_{Cl}$ on (100); (ii) $\theta_{Cl}$ on (111); (iii) $\theta_{HCOO}$ on (100); and (iv) $\theta_{HCOO}$ on (111). Full surface-energy diagram is shown in Supplementary Fig. 20. **f** Changes in $\theta$ of Cl (green) and HCOO (purple) on Pt(100) and Pt(111), respectively, when going from region 2 to region 16 in (**e**). Source data are provided as a Source Data file.

ions in solution: H⁺, K⁺, Cl⁻, and HCOO⁻. Our experiments reveal that the adsorption behavior of these ions is highly sensitive to $A_{tot}$ and that a complex competition between them dictates the surface potential of each facet.

First, we established the relative roles of the positive ions. Comparing HCl and KCl (Fig. 6a), both surfaces are less negatively charged in the presence of H⁺ compared to K⁺, regardless of $A_{tot}$, indicating more H⁺ is adsorbed than K⁺. When half of the K⁺ is replaced with H⁺, this effect was more pronounced on the {100} surface, particularly at high $A_{tot}$ (≥10 m² L⁻¹), where a positive $\triangle\zeta_H$ (the change in $\zeta$ upon replacing half of K⁺ with H⁺) is larger than on the {111} surface (Fig. 6b).

This difference indicates a higher affinity for H⁺ on the {100} surface, which dominates its surface charge in stage ii where {100} OA occurs. In contrast, at low $A_{tot}$ (3 m² L⁻¹), H⁺ has little effect on the change in $\zeta$ for either surface with $\triangle\zeta_H \approx 0$ (Fig. 6b).

Next, we investigated the influence of negative ions. In both HCl and KCl, the {100} surface is more negatively charged than the {111} surface, regardless of $A_{tot}$, indicating greater Cl⁻ adsorption on the {100} surface (Fig. 6a). We then added various amounts of HCOOH in the solution containing 2.9 mM HCl and 2.9 mM KCl. Defining $\triangle\zeta_{HCOO}$ as the change in $\zeta$ upon adding HCOOH (Supplementary Fig. 15, Fig. 6c), at high $A_{tot}$ (≥ 10 m² L⁻¹), we find the addition of HCOO⁻ has a

minimal effect on the change in $\zeta$ for either surface regardless of [HCOOH]. However, at low $A_{tot}$ (3 m² L⁻¹), when [HCOO⁻] exceeds 1.14 mM, the same amount of HCOO⁻ has a dramatic effect, making both surfaces less negatively charged with a $\triangle\zeta_{HCOO}$ of up to 14.3 mV. This suggests there is a complex competition between negatively charged Cl⁻ and HCOO⁻ when the available surface area is scarce.

To directly confirm this competitive adsorption, we used time-of-flight secondary ion mass spectrometry (TOF-SIMS) to measure the surface-bound ions at high and low $A_{tot}$ (Fig. 6d, Supplementary Fig. 18). The analyses were conducted on the {100} and {111} surfaces with $A_{tot}$ of 30 m² L⁻¹ and 3 m² L⁻¹ under the same conditions as in Fig. 5c to qualitatively analyze the relative changes in ion adsorption. As $A_{tot}$ decreases, HCOO⁻ adsorption increases on both surfaces by similar factors. In addition, H⁺ and Cl⁻ adsorption increases on the {100} surface, but decreases on the {111} surface, demonstrating distinct ion competition on the two surfaces. While K⁺ may also participate in competitive adsorption, here we have focused on H⁺ because its adsorption dominates that of the positive ions (Fig. 6a).

Recognizing that the $\zeta$ may be influenced by surface-associated ions within the near-surface interfacial layer, and that charge transfer within that region may impact the actual valency of adsorbed species, a simple comparison of the TOF-SIMS and $\zeta$ data presents the observed crossover in surface potential. The favorable adsorption of H⁺ keeps the {100} surface neutral, allowing for {100} OA in stage ii. As OA repeatedly occurs and $A_{tot}$ decreases (from stage ii to iii), the increase in adsorption of Cl⁻ and HCOO⁻ specifically on the {100} surface overwhelms the H⁺ effect, driving its potential to become strongly negative. Meanwhile, on the {111} surface, the desorption of H⁺ and Cl⁻ partially counterbalance each other, while the adsorption of HCOO⁻ may shift the potential to a slightly more negative value.

To establish a basis for these findings, we performed density functional theory (DFT) calculations with ab initio thermodynamics, which delineates the (100) and (111) surface configurations with the lowest energy as a function of the chemical potential of solution-phase Cl⁻ and HCOO⁻ relative to the gas phase ($\triangle\mu_{Cl^-}$ and $\triangle\mu_{HCOO^-}$) (Fig. 6e, Supplementary Figs. 19 and 20). We note that increasing ion concentration relative to surface area can be equated with increasing $\triangle\mu$ in DFT calculations with a fixed surface area. As expected, at high $\triangle\mu_{Cl^-}$ and low $\triangle\mu_{HCOO^-}$, the surfaces contain only adsorbed Cl[33], while at low $\triangle\mu_{Cl^-}$ and high $\triangle\mu_{HCOO^-}$, only HCOO is adsorbed. The transition from region 2 (low $\triangle\mu$) to region 16 (high $\triangle\mu$) occurs on the borderline between pure HCOO and pure Cl adsorption and represents the shift from stage ii to iii in Fig. 5a. As both values of $\triangle\mu$ increase, the coverages of Cl and HCOO ($\theta_{Cl}$ and $\theta_{HCOO}$) increase on Pt(100), while $\theta_{Cl}$ remains constant and $\theta_{HCOO}$ increases on Pt(111) (Fig. 6f, Supplementary Table 2). These trends align well with the results from $\zeta$ measurements and TOF-SIMS. Thus, DFT results also indicate that branched cube formation occurs in a region of competitive adsorption between HCOO⁻ and Cl⁻ on the Pt surfaces.

The competitive adsorption between Cl and HCOO can be understood in terms of binding energy decreases due to ion–ion repulsion as more ions are adsorbed onto the surface (Supplementary Fig. 21a). Compared to Pt(100), the overall weaker Cl binding energy on Pt(111) (Supplementary Fig. 21a), its significant reduction when Cl co-adsorbs with HCOO (Supplementary Fig. 21b, c), and the relatively smaller co-adsorption regions (Supplementary Fig. 22) indicate HCOO–Cl repulsion is stronger on Pt(111), displacing Cl from Pt(111) at high $\triangle\mu$. On Pt(100), the increasing $\theta_{Cl}$ in the co-adsorption regions (regions 15, 16, 19 to 25 in Supplementary Table 2) further indicates HCOO–Cl repulsion is weaker on Pt(100), assisting Cl to persist on Pt(100) (Fig. 6d, f) under co-adsorption with HCOO, which has a higher binding energy than Cl (Supplementary Table 2, Supplementary Fig. 23), at high $\triangle\mu$.

## Discussion

OA has important implications for achieving specific material properties because it allows NPs to spontaneously organize, enabling control over size and morphology that cannot be achieved through ion-by-ion crystal growth. The driving force behind OA is the net reduction in free energy that comes about with the reduction in surface area achieved when particles merge. However, minimization of free energy alone cannot account for the facet-selective nature of attachment, as interacting NPs cannot inherently distinguish between surfaces with higher or lower energy, and attachment on any matched lattice plane will reduce the free energy, even if attachment on one specific set of planes reduces it the most. Thus, without a source of torque, when particles interact with random orientations driven by Brownian motions, facet-selectivity is only possible if either the barriers to attachment on all other planes are too large to overcome, or attachment is a reversible process, allowing the particles to sample all possible configurations and thus find the lowest energy plane of attachment. Even then, there would be a distribution of attachment planes that reflect the relative changes in free energy. Neither of these situations is present in the Pt NP system.

The solution to this puzzle lies in physical torques that rotate and align the NPs against inherent Brownian torques. While torques created by inherent interparticle interactions, such as vdW and dipole–dipole, are known to contribute to this process (Supplementary Note 2)[10,29–31], the findings presented here now extend this concept, identifying anisotropic electrostatic interactions as a new, externally tunable source of such torques (unlike vdW and dipole–dipole torques, which originate from the intrinsic properties of the materials). Moreover, the mechanism of using competitive ion adsorption to generate tunable electrostatic torques is not limited to the Pt system but represents a generalizable strategy for rationally designing complex nanomaterials, particularly when pH is recognized as another means for manipulating surface potentials through H⁺ and OH⁻ adsorption, because this strategy relies on two ubiquitous features of crystalline materials: the existence of distinct facets and the facet-dependent nature of their surface chemistry. Thus, this approach can be readily extended to other noble metals, oxides, and semiconductors, where ion- or pH-sensitive charge of surface terminations provides a natural handle for electrostatic manipulation[19,34–36]. While this approach may seem to be inapplicable to cubic systems or any system that is characterized by growth habits for which the faces are crystallographically identical, that is not the case, because all crystallographic directions are represented nonetheless, even if only at rounded corners. Recent results on the growth of hematite (Fe₂O₃) by OA show that crystals with very different growth habits all attach along the <001> direction, even when there is no (001) facet present, and thus attachment must occur on the corners of the nanocrystals[9].

The collective findings in this study demonstrate that the interplay of ion adsorption and resultant surface potential can appreciably influence attachment behavior during OA. Anisotropic electrostatic interactions, created by competing ions with facet-dependence, induce approaching NPs to preferentially orient before attachment. This surface potential evolves dynamically in response to changes in the relative surface coverages of the competing ions as surface area decreases during repeated OA events, potentially switching the direction of OA. Unlike intrinsic vdW and dipole–dipole interactions, electrostatic interactions can be finely and dynamically tuned across all surfaces by adjusting solution chemistry, such as pH, electrolyte type, and concentration. While electrostatic interactions have long been recognized as important in NP assembly, their role has typically been considered in terms of static attractive or repulsive forces rather than as dynamic, facet-specific torques capable of switching attachment pathways in real time. This suggests that the dynamic manipulation of these facet-specific electrostatic interactions and resultant torques

provides an underexplored strategy for engineering NP assemblies, enabling precise spatial and temporal control over the OA process.

## Methods

### Materials

Potassium tetrachloroplatinate(II) ($K_2PtCl_4$, 98%), formic acid (HCOOH, 99%), potassium chloride (KCl), and hydrochloric acid (HCl, ACS reagent, 37%) were purchased from Sigma-Aldrich.

### Growth of branched cubic Pt mesocrystals

For a typical synthesis, we dissolved 6 mg $K_2PtCl_4$ (Pt precursor) in 10 mL distilled water and added 20 μL formic acid (reducing agent) under constant stirring, yielding final concentrations of 1.45 mM $K_2PtCl_4$ and 52 mM formic acid. The solution was settled at room temperature, changing from clear yellow to opaque dark brown at ≈10 min, which indicates Pt NPs formation (Supplementary Fig. 2), and eventually turning opaque dark gray. Conditions with various reactant concentrations are presented in Supplementary Table 1.

### In situ LPTEM experiments

We performed LPTEM using an Insight Chips® nanochannel holder (130 nm channel height) with silicon nitride encapsulation (25 nm thickness) coated on the inner surfaces with $Al_3O_4$[21,22]. The Pt precursor solution was loaded into the nanochannel, and diluted formic acid was injected through another inlet. A FEI (a subsidiary of Thermo Fisher Scientific) Tecnai F20 microscope was operated at 200 kV with a Gatan OneView IS camera, and movies were recorded at 10 fps with an electron beam dose rate of 6 e⁻ Å⁻² s⁻¹. The movies were processed as a 3-frame averaged time series using a Gatan DigitalMicrograph.

For the experiment in Supplementary Fig. 7, we used a Hummingbird Scientific liquid holder with a 250 nm Au spacer between two square window chips with silicon nitride membranes. The membranes were oxygen plasma cleaned for 2 min (Harrick Plasma) to render them hydrophilic prior to use. A 0.3 μL droplet of the reaction solution was loaded onto the spacer chip, covered with the second chip. The assembled liquid holder was placed in a pump station (Pfeiffer Vacuum) to check the vacuum status before loading into TEM. The dose rate was maintained at 10 e⁻ Å⁻² s⁻¹.

### Cryo-TEM experiments

Cryo-TEM was performed on a 200-kV FEI (a subsidiary of Thermo Fisher Scientific) Tecnai F20 microscope and JEOL GrandARM-300F, both equipped with a Gatan OneView IS camera. A Krios G3i cryo-TEM was used for low-dose and HR-cryo-TEM observations. At different time points, 3 μL of reaction solution was applied to a lacey carbon film Cu TEM grid and vitrified by blotting for 4 s and pluge-freezing in liquid ethane using an FEI (a subsidiary of Thermo Fisher Scientific) Vitrobot Mark III. The TEM grid was glow-discharged for 25 s at 15 mA using PELCO easiGlow™ prior to use. Images were obtained with a defocus value between −1 and −2 μm, and an accumulated total dose below 200 e⁻ Å⁻². Note that intermediates—solvated NPs, clusters, and core/shell structures—were observable only in the solution phase, when the NPs are in a solvated state (Supplementary Fig. 10).

### S/TEM experiments

For S/TEM analysis, a drop of the sample solutions was loaded onto a pure carbon film Cu TEM grid. Conventional TEM images were acquired at 300 kV on a FEI (a subsidiary of Thermo Fisher Scientific) Titan Environmental TEM with a Gatan UltraScan 1000 camera and JEOL GrandARM-300F with a Gatan OneView IS camera. Scanning TEM (STEM) images were acquired in STEM mode on a JEOL GrandARM-300F.

### SAED pattern analysis

The azimuthal profile and d-spacing of the SAED pattern were analyzed using a Gatan Digital Micrograph with the PASAD script.

### SEM experiments

SEM imaging was performed on Carl Zeiss XB.

### pH measurements

All pH measurements were conducted with a Mettler Toledo pH meter.

### Estimation of ion concentrations before and after Pt reduction

Calculations are based on a solution containing 1.45 mM $K_2PtCl_4$ and 52 mM HCOOH. Before the reaction, equilibrium concentrations based on the equilibrium constant ($K_a$) of formic acid are [HCOOH] = 49.03 mM, [HCOO⁻] = [H⁺] = 2.97 mM. The calculated pH of 2.53 is consistent with the measured value of 2.52 ± 0.04 (averaged over 10 measurements).

The reaction is:

$$K_2PtCl_4 + HCOOH \rightarrow Pt + 2K^+ + 2H^+ + 4Cl^- + CO_2.$$

Assuming complete reduction of 1.45 mM $K_2PtCl_4$, the ion concentrations are adjusted as follows: [K⁺] = 2.9 mM, [Cl⁻] = 5.8 mM, [H⁺] = 2.97 + 2.9 = 5.87 mM, [HCOOH] = 49.03 − 1.45 = 47.58 mM, and [HCOO⁻] = 2.97 mM. Subsequently, the concentrations of H⁺, HCOO⁻, and HCOOH are re-equilibrated based on the $K_a$ of formic acid as follows:

$$\frac{([H^+]+x)([HCOO^-]+x)}{[HCOOH]-x} = K_a, \tag{1}$$

yielding final concentrations of [H⁺] = 4.75 mM, [HCOOH] = 48.7 mM, and [HCOO⁻] = 1.85 mM. The final calculated pH of 2.32 closely matches the measured pH of 2.36 ± 0.01 after 1 h, suggesting that most of the $K_2PtCl_4$ has been reduced under the synthesis conditions of branched cubic mesocrystals.

### UV-vis measurements and NP concentration estimation

UV-vis absorbance spectra were obtained using a UV-vis UV-2600 Spectrophotometer (Shimadzu Scientific) with a quartz cuvette against a distilled water background. The spectra for 1.45 mM $K_2PtCl_4$ and 52 mM formic acid, respectively, are shown in Supplementary Fig. 13a[37,38]. Based on the Beer-Lambert law, the concentration of Pt NPs can be qualitatively estimated from the intensity at 236 nm of wavelength, assuming that the intensity change at 236 nm was associated with the intensity change at <200 nm, which is the absorption peak of Pt NPs (Supplementary Fig. 13, Fig. 5b)[39]. The spectrum was identical to that of $K_2PtCl_4$ for ≈9 min, indicating no significant Pt reduction had occurred. The intensity at 236 nm began to increase after ≈10 min, reached its maximum at ≈20 min, and then gradually decreased, showing little change after ≈40 min.

The concentration of Pt NPs in solution was calculated using

$$C = \frac{A}{e^{10.60}D^{2.84}}, \tag{2}$$

which is derived from the relationship between the extinction coefficient for Pt and the average particle size[40]. $C$ is the concentration of NPs (mol L⁻¹), $A$ is the absorbance of a solution at 550 nm, e is the Euler's number, and $D$ is the average particle diameter (nm).

The calculation of the initial NP concentration was based on a 3 nm truncated octahedron consisting of 586 atoms, a representative equilibrium configuration of small NPs[26,41]. From an initial 1.45 mM of $K_2PtCl_4$, the concentration of these 3 nm NPs is calculated to be 2.5 μM, as follows:

$$C = \frac{1.45 \frac{mmol}{L} \times N_A \frac{atoms}{mol}}{586 \frac{atoms}{NP} \times N_A \frac{NP}{mol}} = 2.5\,\mu M. \tag{3}$$

After these primary NPs aggregate into 120 nm cubes, the final concentration is ≈50 pM, which is analogous to the 58 pM calculated from UV-vis absorbance at 550 nm after 2 days of reaction.

## Synthesis of Pt cubes and octahedra

NPs were synthesized with slight modifications to methods from refs. [42,43]. For Pt octahedra, 2.5 mL of ethylene glycol (EG) was refluxed for 5 min in a 25 mL round-bottom flask under magnetic stirring at 180 °C in an oil bath. Then, 0.5 mL of 0.06 M silver nitrate in EG was added to the boiling EG. Over a 16-min period, 3 mL of 0.375 M poly(vinylpyrrolidone) ($M_w$ 29 000 g mol$^{-1}$) and 1.5 mL of 0.0625 M $K_2PtCl_4$ in EG were added dropwise in 32 equal portions, every 30 s. The solution was then refluxed for an additional 5 min, resulting in a dark brown solution.

For Pt cubes, 0.05 mmol $H_2PtCl_6 \cdot 6H_2O$, 0.75 mmol of tetramethylammonium bromide, and 1.00 mmol of poly(vinylpyrrolidone) ($M_w$ 29 000 g mol$^{-1}$) were dissolved in 10 mL of EG in a 25 mL round-bottom flask at room temperature, resulting in a clear yellow solution. The mixed solution was then heated in an oil bath at 180 °C and refluxed for 40 min under magnetic stirring, resulting in a dark brown solution.

After cooling to room temperature, a triple volume of acetone was added to the solution, followed by centrifugation at 1650 × g for 5 min to collect the precipitate. The product was then washed three times by redispersing in a 12 mL of ethanol/hexane mixture (1:3 v/v) and centrifuging at 1650 × g for 5 min.

## Dynamic light scattering and zeta potential measurements

A 10 mL of reaction solution including 1.45 mM $K_2PtCl_4$ and 52 mM formic acid was prepared, and a 1 mL of reaction solution was loaded into a quartz cuvette. Hydrodynamic size and zeta potential were measured on a Zetasizer Nano ZS (Malvern) with a Dip cell kit. Zeta potential of pre-synthesized Pt NPs with predominately {100} and {111} faces (Supplementary Fig. 14) were characterized using a Lite-sizer 500 DLS (Anton Paar) with a polystyrene cell. Dispersions were ultrasonicated for 2 min and immediately measured. Polystyrene latex beads were used for instrument validation before measurement.

## TOF-SIMS measurements

Pt cube and octahedron NPs were dispersed in a solution containing 2.9 mM HCl, 2.9 mM KCl, and 26.5 mM HCOOH, then sonicated for 5 min and centrifuged at 12,100 × g for 10 min. After removing the supernatant, ≈1 mL ultrapure water (18.2 MΩ cm$^{-1}$) was added, and the solution was centrifuged again. After removing the water, the precipitates were placed on a clean Si wafer and dried under ambient conditions for analysis. Prior to use, the Si wafer was cleaned by sequential ultrasonication in acetone, isopropanol, and ultrapure water (5 min each), followed by drying with pure $N_2$ gas and a 1 min UV-ozone treatment to make the surface hydrophilic.

TOF-SIMS surface spectra were acquired using an instrument from IONTOF GmbH. A 25 keV Bi$^+$ analysis beam with a beam current of 1.10 pA at 10 kHz was focused to a 5 μm diameter and scanned over a 300 × 300 μm$^2$ or 500 × 500 μm$^2$ area. Measurements were taken at 4-6 locations on each sample. Due to matrix effect, direct quantification of adsorbed ions is difficult, therefore, based on a widely-used semi-quantification strategy, we normalized the signal intensity of each desirable species to that of Pt. This relative intensity ratio is used to represent the change in ion adsorption between $A_{tot}$ of 3 m$^2$ L$^{-1}$ and 30 m$^2$ L$^{-1}$, qualitatively explaining the change in surface potential. In the negative ion spectrum, the intensity ratio of Pt with adsorbates (Cl, HCOO, and H) to pure $^{194}$Pt was calculated for a rough comparison of adsorption under each condition. Because of the overlap of $^{194}$PtH$^-$ to $^{195}$Pt$^-$ at $m/z$ 195, $^{194}$PtH$^-$ intensity, which can represent H adsorption, can be calculated by removing the

contribution of $^{195}$Pt$^-$ based on the isotope ratio of $^{194}$Pt (32.9%) to $^{195}$Pt (33.8%). To avoid interference from various isotopes, mass regions of interest (ROIs) corresponding to $^{194}$Pt$^{35}$Cl ($m/z$ 228.93), $^{194}$PtH$^{12}$C$^{18}$O$_2$ ($m/z$ 238.97), and $^{195}$Pt + $^{194}$PtH ($m/z$ 194.97) were used to calculate the adsorption ratios for Cl, HCOO, and H on Pt surfaces.

High peaks related to Ag were detected in the octahedron NP samples, presumed to be residual from the AgNO$_3$ used during synthesis. We concluded that Ag$^+$ adsorption does not affect the overall surface charge or the distinctive nature of charge between {100} and {111} surfaces based on three observations. (1) The octahedron NPs are negatively charged overall, similar to NPs in the synthesis condition of branched cubic mesocrystals (Supplementary Fig. 15). (2) The {100} facets show a more negative zeta potential than the {111} facets in pure HCl or pure KCl (Fig. 6a), consistent with DFT results showing higher Cl$^-$ coverage ($\theta$) on the (100)[33]. (3) In pure HCOOH, the {100} facets show a similar zeta potential to the {111} facets but are slightly more negatively charged (Supplementary Fig. 17), consistent with DFT results for HCOO$^-$ on two surfaces (Supplementary Fig. 23). These results align with the DFT finding that Cl$^-$ and HCOO$^-$ preferentially adsorb on the (100) surfaces. Therefore, the residual Ag may not significantly influence facet-dependent ion adsorption and zeta potential.

## Computational methods

All DFT calculations were performed using the Vienna Ab initio Simulation Package (VASP)[44–46] with projector augmented waves (PAW)[47]. The generalized gradient approximation (GGA) by Perdew, Burke, and Ernzerhof (PBE) was used for the exchange-correlation functional[48]. We chose an energy cutoff of 450 eV as an optimal value for our plane-wave basis set. For sampling the first Brillouin zone, Monkhorst-Pack grids were used[49]. The unit cells with corresponding $k$-point meshes are listed in Supplementary Table 3. We also included the DFT-D3 method of Grimme with the Becke-Jonson (BJ) damping to describe long-range vdW interactions[50].

A (15 × 15 × 15) $k$-point grid was used to optimize and calculate the energy of bulk Pt. We found a bulk Pt lattice parameter of $a = 3.92$ Å. A single $k$ point was used to compute energies of gas-phase Cl and HCOO in a unit cell box with $a = b = c = 20$ Å. To investigate the adsorption of solution-phase Cl$^-$ and HCOO$^-$ on Pt(100) and Pt(111), we used a periodic slab that has six layers. We fixed the bottom three layers to realize bulk positions. We included a vacuum spacing of 15 Å and dipole correction in the $z$-direction normal to the surface for calculations involving adsorbates on the Pt surfaces, to prevent unphysical interactions between periodic cells. The energy convergence criterion of 10$^{-6}$ eV and force convergence criteria of 0.05 eV Å$^{-1}$ were used for all geometry optimizations. By adding HCOO to the Cl adsorption configurations on Pt(100) and Pt(111) obtained in our previous study, we delineated the configurations with the lowest surface energy as a function of the chemical potentials (△$\mu$) of Cl$^-$ and HCOO$^-$ on the Pt surfaces relative to the gas-phase chemical potentials (Fig. 6e, Supplementary Figs. 19 and 20)[33]. The configurations we observe at very low △$\mu_{HCOO^-}$ are consistent with ref. [33]–though here we use higher $k$-point densities than in our previous work. To assess the competitive adsorption of HCOO$^-$ and Cl$^-$ on the Pt surfaces, we studied three quantities: surface energy ($\gamma$), coverage ($\theta$), and binding energy ($E_{bind}$) of each adsorbed species (Supplementary Table 2). In ab initio thermodynamics analysis of the adsorption of chemical species on Pt surface, the surface energy, $\gamma_{Pt-Cl-HCOO}$ was calculated using

$$\gamma_{Pt-Cl-HCOO} = \frac{E_{Pt-Cl-HCOO} - N_{Pt}E_{Pt}^{Bulk} - N_{Cl}\mu_{Cl^-} - N_{HCOO}\mu_{HCOO^-}}{A_{surf}} - \gamma_{Pt}^{fixed}. \quad (4)$$

The binding energy ($E_{bind}$) of each species was computed using

$$E_{bind, Cl} = \frac{\left[(E_{Pt-HCOO} + N_{Cl}E_{Cl}) - E_{Pt-Cl-HCOO}\right]}{N_{Cl}}, \quad (5)$$

and

$$E_{bind, HCOO} = \frac{\left[(E_{Pt-Cl} + N_{HCOO}E_{HCOO}) - E_{Pt-Cl-HCOO}\right]}{N_{HCOO^-}}. \quad (6)$$

Here, $E_{Pt-Cl-HCOO}$ is the energy of an optimized Pt-Cl-HCOO slab, $N_{Cl}$ is the number of Cl on the Pt surface, $N_{HCOO}$ is the number of HCOO on the Pt surface, $E_{Pt-HCOO}$ and $E_{Pt-Cl}$ are the energies of the Pt-Cl-HCOO slabs with Cl and HCOO removed, respectively. $A_{surf}$ is a surface area of Pt surface, and $\gamma_{Pt}^{fixed}$ is the computed energy for a Pt slab with all Pt atoms fixed. When $E_{bind, x}$ is positive, the adsorption of species $x$ is thermodynamically favorable.

### Reporting summary
Further information on research design is available in the Nature Portfolio Reporting Summary linked to this article.

## Data availability
The data that support the findings of this study are available from the corresponding authors upon request. TOF-SIMS surface spectra are available from Figshare[51]. Source data are provided with this paper.

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

## Acknowledgements

We acknowledge Xinran Liang for his assistance with the inset graphic of Fig. 1a. This research was performed at Pacific Northwest National Laboratory (PNNL) and was initially supported by the US Department of Energy (DOE) Office of Science (SC) Basic Energy Sciences (BES) Division of Materials Science and Engineering (MSE) Synthesis and Processing Sciences program (SPSP) under Early Career Research Program award FWP 67037 (D.L., H. Zhang, and J.H.), before transitioning to Award FWP 67554 (J.J.D., D.L., J.C., Y.B., and J.H.). Initial chemical synthesis, SEM, TEM characterization, and in situ TEM work were supported by Early Career Award FWP 67037. HR-cryo-TEM, in situ TEM, UV-Vis, zeta potential, and chemical synthesis were supported by FWP 67554. DFT calculations were funded by the DOE BES MSE SPSP, Grant DE-FG02-07ER46414 (K.A.F. and E.K.). HR-cryo-TEM, environmental TEM, UV-vis, and TOF-SIMS were performed on a project award (60753) from the Environmental Molecular Sciences Laboratory (EMSL), a DOE Office of Science User Facility sponsored by the Biological and Environmental Research program under Contract No. DE-AC05-76RL01830. Pacific Northwest National Laboratory is a multiprogram national laboratory operated for the DOE by Battelle under Contract DE-AC05-76RL01830. DFT calculations used Bridges-2 at the Pittsburgh Supercomputing Center through allocation DMR110061 from the Advanced Cyberinfrastructure Coordination Ecosystem: Services & Support (ACCESS) program, which is supported by National Science Foundation grants #2138259, #2138286, #2138307, #2137603, and #2138296. Assistance with the surface analysis was supported by the U.S. DOE Office of Science-Basic Energy Sciences, under Contract No. DEAC02-06CH11357 (H. Zhou).

## Author contributions

Y.B. conducted experiments and data analysis. E.M.K. conducted DFT calculations. K.A.F. supervised DFT calculations. J.C. developed the theoretical formulation and assisted with the theoretical calculation. Z.Z. performed TOF-SIMS and assisted with the analysis. H. Zhang performed the initial synthetic experiments and in situ TEM study. J.H. assisted with the initial zeta potential measurement. Y.K.S. and H. Zhou assisted with the analysis of surface chemistry and structure from modeling and experimental aspects, respectively. T.H.M. assisted with HR-cryo-TEM. J.E.E. provided oversight with the HR-cryo-TEM data collection from Krios TEM. E.C.S.J. and K.S.M. assisted with LPTEM. Y.B., D.L., and J.J.D.Y. wrote the manuscript. E.M.K. and K.A.F. contributed to writing the DFT section. Y.B., D.L., J.J.D.Y., E.M.K., K.A.F., J.C., and K.S.M. revised the manuscript. D.L. designed the experiments and K.A.F. designed the simulations. D.L. and J.J.D.Y. supervised the research. K.A.F. supervised the simulations. All authors contributed to the discussion of results and reviewed the manuscript.

## Competing interests
