## [Transparent Peer Review file · Nature Communications]

Anisotropic surface potentials induced by competitive ion adsorption enable the synthesis of branched cubic Pt mesocrystals

Corresponding Author: Dr Dongsheng Li

Version 0:

Reviewer comments:

Reviewer #1

(Remarks to the Author)

The manuscript presents a detailed investigation into oriented attachment mechanisms in Pt nanoparticle systems, with particular emphasis on facet-specific electrostatic interactions and the role of ion adsorption in modulating surface potentials.

I personally find the central idea -- that anisotropic electrostatic interactions, complemented by shape anisotropy, can induce nanoparticles to assemble along preferential directions -- both fascinating and conceptually rich. However, I have concerns regarding the accessibility of the manuscript. Even as a researcher in a related area, I found the paper extremely dense, both in its technical language and conceptual presentation. Key figures (especially Figure 1) are difficult to interpret without deep familiarity with the system and the methodologies involved. Moreover, the manuscript lacks a clear contextual framing of the problem within the current landscape of related research, which further limits its accessibility and fails to convincingly demonstrate its novelty and timeliness.

In my opinion, the current presentation makes this work more suitable for a specialist journal aimed at a narrower and technically expert readership. Unless the manuscript is substantially revised to enhance clarity, contextualization, and accessibility, I do not consider it appropriate for publication in Nature Communications, which addresses a broader materials science audience.

Reviewer #2

(Remarks to the Author)

This is a thoroughly researched work that utilizes an extensive battery of modern experimental techniques. The research has a great internal logic, despite the complexity of the problem it deals with. I like the article.

The understanding of the morphogenesis of crystalline materials created by the assembly of nanoparticles through oriented aggregation - also referred to as mesocrystals by some research groups - is typically explained by forces between the particles, including electrostatic repulsion, which can be tuned by solution chemistry. The authors claim that anisotropies in surface potential, driven by competitive ion adsorption, can alter the selection of faces during the oriented aggregation of nanoparticles.

The authors perform experiments varying the ratio of reactants and adding either KCl or HCl, and follow the effect on morphology. The results are clear, and I agree with the explanation provided by the authors well supported by TEM and LPTEM videos.

However, the main experiment of the authors, as written in the section "Synthesis and structural characterization of branched Pt cubes", describes the generation of two consecutive types of oriented aggregation, namely the growth of a cubic core through {100} attachment and later at 40-45 minutes the extension of {111}-oriented branches out from the cubic core through {111} attachment. According to the experimental description, it does not appear that there is any change in the solution chemistry during the experiment beyond that caused by the formation of Pt particles, which is negligible. In fact, as shown in Figure 4a, neither the pH nor the UV absorbance changes significantly during the experiment and remain almost

constant at 40-45 minutes, when the morphological transition is recorded. The hypothesis that competitive ion adsorption can explain the transition is, in my opinion, an ad hoc hypothesis. It could be accepted in the best of the cases for this particular experiment. To make the explanation universal, I strongly suggest that the authors compare the morphogenetic power of ion adsorption with that of solution composition and the solution chemistry independent dipole–dipole interactions and anisotropic van der Waals (vdW) interactions.

The videos are very informative. I suggest adding a scale bar to the videos, as it is currently only displayed at the caption of the videos, which is not very practical. Also, use real-time instead of 4x. Currently, they last only a few seconds, making it difficult to appreciate the information they provide fully.

As I have already said, the work is of a great technical quality. However, the authors should make an effort to explain the importance and novelty of their work, i.e. the importance of competitive ion adsorption in controlling OA beyond this case of platinum nanoparticles aggregation.

Reviewer #3

(Remarks to the Author)

In this manuscript the authors investigate how ion adsorption can induce time-dependent, facet-specific changes in surface potential that govern the oriented attachment and morphology of Pt mesocrystals. They combine direct observations from LP-EM and cryo-TEM, with mechanistic interpretation following zeta potential measurements, TOF-SIMS, and DFT calculations. The authors show branched Pt NP grow through two stages of oriented attachment which can be manipulated through dynamic, facet-specific changes in surface potential via adsorption of Cl^- , H^+ , and HCOO^- ions. The authors claim this strategy can be generally applied to the synthesis of complex nanostructures.

The study is thorough; the methodology is sound and offers a detailed mechanistic understanding of forces governing Pt NP assemblies.

A few comments that should be addressed prior to publication:

1. The landscape of previous works on material synthesis through OR is not detailed enough in the introduction, making it difficult to grasp the novelty of the current work easily. Specific examples of findings from previous studies like Liu, Nat Commun 11, 1045 (2020) (where the relationships between structure, forces and response dynamics in OA of ZnO NP was established) could help.
2. The following statement is too general “However, manipulating the OA process to produce novel architectures at will, based on a fundamental understanding of the underlying interparticle forces has not been possible.” Examples of previous works: (1) Biao Jin, Angew Chemie 16, e202201980 (2022), (2) Sushko, Nanoscale 8, 19714-19725 (2016)
3. Could the switch between the growth of cubic core {100} and branching extensions {111} be visualized by LP-EM? Could the authors explain their rationale for showing the later stages of mesocrystal formation by cryo-TEM and not LP-EM? This should be addressed in the text.
4. I find Supplementary Fig. 11 to be very helpful in visualizing the role of the different species on the morphology. The authors should consider moving it from the SI to the main text, perhaps as part of Fig. 4.
5. In the discussion the authors state “the action of electrostatic torques must be a general phenomenon and may govern the facet selection of OA in a wide range of NP systems”, could the authors explain more specifically how this can be done? As switching mechanism relies on specific ion adsorption behaviours on Pt facets, which may not extend to other materials. Also, the statement “This suggests that the ability to manipulate electrostatic interactions and resultant electrostatic torques provides an unexplored strategy for engineering NP assemblies with spatial and temporal control of the OA process.” seems overstated. While specifically using ion-modulated, facet-specific electrostatic torques to trigger dynamic facet switching in OA is novel, the broader concept of spatial and temporal control of nanoparticle assembly through electrostatic manipulation is not.

Minor comments:

6. In Fig. 4d and e, it would be helpful to draw the reader's attention to the change in axis or keep the axis the same in both so that the different trend is clearly visible.
7. In Fig. 1e,g circles are drawn around the NPs making up the mesocrystal. I find it difficult to differentiate the individual NP from the image, can the authors annotate the images in a way that the reader can easily identify these?
8. Some supplementary figures are not referred to correctly in the main text (example: Supplementary Fig. 15 should be 16). Please check this during revision.

Version 1:

Reviewer comments:

Reviewer #1

(Remarks to the Author)

I am satisfied with the changes made by the author. The manuscript has been significantly and thoroughly revised, with added references and enhanced context that greatly improve its accessibility for a broader audience. Additionally, the schematic diagrams of the assemblies are very informative and contribute to the clarity of the work. Overall, I believe the

paper is now ready for publication and recommend giving it the green light.

Reviewer #3

(Remarks to the Author)

The authors have addressed all my comments well and improved the manuscript, providing much needed context and explanations. The work is of high quality and significance.

Responses to the reviewers' comments

We truly appreciate the reviewers' valuable comments on our manuscript (NCOMMS-25-38790-T), entitled "Anisotropic surface potentials induced by ion adsorption enable synthesis of complex nanostructures". We have carefully addressed all criticisms in the revised manuscript, as indicated in our responses to specific reviewers' comments. Reviewers' comments are in italics, and our responses follow in plain text. Our revisions are listed in bullets (●) after the response and modifications in the main text are in blue.

Reviewer 1

General comments: *The manuscript presents a detailed investigation into oriented attachment mechanisms in Pt nanoparticle systems, with particular emphasis on facet-specific electrostatic interactions and the role of ion adsorption in modulating surface potentials.*

*I personally find the central idea -- that anisotropic electrostatic interactions, complemented by shape anisotropy, can induce nanoparticles to assemble along preferential directions -- both fascinating and conceptually rich. However, I have concerns regarding the accessibility of the manuscript. Even as a researcher in a related area, I found the paper extremely dense, both in its **technical language and conceptual presentation**. Key figures (especially **Figure 1**) are difficult to interpret without deep familiarity with the system and the methodologies involved. Moreover, the manuscript lacks a **clear contextual framing** of the problem within the current landscape of related research, which further limits its accessibility and fails to convincingly demonstrate **its novelty and timeliness**.*

*In my opinion, the current presentation makes this work more suitable for a specialist journal aimed at a narrower and technically expert readership. Unless the manuscript is substantially revised to enhance **clarity, contextualization, and accessibility**, I do not consider it appropriate for publication in Nature Communications, which addresses a broader materials science audience.*

Response: We appreciate the reviewer's thoughtful and highly constructive feedback on our manuscript. We agree that the original version was dense and did not sufficiently frame our work for the broad readership of Nature Communications. The reviewer's comments have been invaluable in guiding us to substantially revise the manuscript to enhance its clarity, contextualization, and accessibility. We believe the manuscript is now significantly improved.

- We revised the introduction as follows to provide a comprehensive overview of the current landscape of Oriented Attachment (OA) research. We detail the progress made in understanding the interparticle forces at play and then clearly define the specific, unresolved challenges in the field. This new framing allows us to more convincingly situate our work and demonstrate how it addresses a critical question regarding the dynamic control of nanocrystal assembly.

Revised text in the introduction (Page 2, Line 2; in the main text):

"When crystalline materials form through assembly of nanoparticles (NPs), their properties are strongly influenced by the assembled architecture because phenomena like photon and electron scattering, electron-hole recombination, and dislocation generation depend on the

material's characteristic length scales and topology¹⁻³. Oriented attachment (OA) has emerged as a key pathway for creating single-crystal-like structures with diverse morphologies⁴⁻⁷.

During OA, neighboring particles align and fuse along matching crystallographic planes, but intriguingly, they often do so with high facet-selectivity—certain crystallographic facets are repeatedly favored over others^{4,8,9}. This consistent selectivity is surprising because particles in solution constantly undergo random Brownian motion, leading to sampling of all possible orientations. Why particles consistently select specific crystallographic facets when attachment on any set of matched lattice planes would reduce the system's energy remains an unresolved question in the field. Answering that question would address one of the key challenges in designing crystal structures: understanding how to control and select specific facets for OA, thus enabling the tailoring of OA-based crystal growth and assembly.

Recent efforts have begun to unravel the complex interplay of interparticle forces that drive OA. For example, real-time imaging of OA in the ZnO system has established a relationship between particle structure, interaction forces arising from ion-solvent correlations and dipolar interactions, and the resulting assembly dynamics¹⁰. Other studies in metal oxide systems have explored the influence of ion-correlation forces on OA¹¹. These studies show that OA kinetics and pathways are intricately tied to interfacial chemistry, which is governed by environmental factors, including electrolyte type and concentration, surface adsorbates, and pH¹²⁻¹⁷. However, despite this progress, current understanding remains largely focused on interaction energetics, with little attention paid to how directional rotational alignment is achieved. Since facet-selective OA requires precise orientation at the moment of collision, this gap points to the need for a mechanism capable of generating directional torque, especially at nanometer distance.

Amongst the forces defining interparticle potentials, the repulsive electrostatic force is most strongly dependent on the chemistry of the NP surface as it relies on the surface potential, which can vary independently on different crystal facets^{18,19}. Here we show that the facet-specificity of attachment by Pt NPs can be manipulated through changes in the surface potential of distinct facets to drive a transition from attachment on Pt{100} to Pt{111}. This transition leads to a switch from growth of a cubic core through {100} attachment to the extension of {111}-oriented branches out from the cubic core through {111} attachment. Moreover, we find that the disparity in the surface potential between the two facets creates an electrostatic torque that is critical for ensuring facet specificity. Finally, we demonstrate that competitive ion adsorption underlies the changes in surface potential that leads to the transition from {100} to {111} attachment through torque-driven facet selection.”

8. Pacholski, C., Kornowski, A. & Weller, H. Self-Assembly of ZnO: From Nanodots to Nanorods. *Angewandte Chemie International Edition* **41**, 1188–1191 (2002).
9. Wang, Y. *et al.* Particle-based hematite crystallization is invariant to initial particle morphology. *Proceedings of the National Academy of Sciences* **119**, e2112679119 (2022).
10. Liu, L. *et al.* Connecting energetics to dynamics in particle growth by oriented attachment using real-time observations. *Nat Commun* **11**, 1045 (2020).

11. Sushko, M. L. & Rosso, K. M. The origin of facet selectivity and alignment in anatase TiO₂ nanoparticles in electrolyte solutions: implications for oriented attachment in metal oxides. *Nanoscale* **8**, 19714–19725 (2016).

17. Jin, B. *et al.* Peptoid-Directed Formation of Five-Fold Twinned Au Nanostars through Particle Attachment and Facet Stabilization. *Angew Chem Int Ed* **61**, e202201980 (2022).

- We revised the discussion as follows to better illustrate the significance and broader implications of our findings. We added a dedicated section on the generalizability of the "competitive ion adsorption" mechanism, explaining how this principle can be applied to other important material systems beyond platinum.

Revised text in the discussion (Page 7, Line 30; in the main text):

“OA has important implications for achieving novel material properties, because it allows NPs to spontaneously organize, enabling control over size and morphology that cannot be achieved through ion-by-ion crystal growth. The driving force behind OA is the net reduction in free energy that comes about with the reduction in surface area achieved when particles merge. However, minimization of free energy alone cannot account for the facet-selective nature of attachment, as interacting NPs cannot inherently distinguish between surfaces with higher or lower energy, and attachment on any matched lattice plane will reduce the free energy, even if attachment on one specific set of planes reduces it the most. Thus, when particles interact with random orientations driven by Brownian motions, facet-selectivity is only possible if either the barriers to attachment on all other planes are too large to overcome, or attachment is a reversible process, allowing the particles to sample all possible configurations. Even then, there would be a distribution of attachment planes that reflect the relative changes in free energy. Neither of these situations is present in the Pt NP system.

The solution to this puzzle lies in physical torques that rotate and align the NPs against inherent Brownian torques. While torques created by inherent interparticle interactions, such as vdW and dipole–dipole, are known to contribute to this process (Supplementary Note 8)^{10,29–31}, the findings presented here now extend this concept, identifying anisotropic electrostatic interactions as a new, powerful, and externally tunable source of such torques (unlike vdW and dipole–dipole torques, which originate from the intrinsic properties of the materials). Moreover, mechanism of using competitive ion adsorption to generate tunable electrostatic torques is not limited to the Pt system but represents a generalizable strategy for rationally designing complex nanomaterials, particularly when pH is recognized as another means for manipulating surface potentials through H⁺ and OH[−] adsorption, because this strategy relies on two ubiquitous features of crystalline materials: the existence of distinct facets and the facet-dependent nature of their surface chemistry. Thus, this approach can be readily extended to other noble metals, oxides, and semiconductors, where ion- or pH-sensitive charge of surface terminations provides a natural handle for electrostatic manipulation^{19,34–36}. While this approach may seem to be inapplicable to cubic systems or any system that is characterized by growth habits for which the faces are

crystallographically identical, that is not the case, because all crystallographic directions are represented nonetheless, even if only at rounded corners. Recent results on growth of hematite (Fe_2O_3) by OA show that crystals with very different growth habits all attach along the $\langle 001 \rangle$ direction even when there is no (001) facet present and thus attachment must occur on the corners of the nanocrystals⁹

The collective findings in this study demonstrate that the interplay of ion adsorption and resultant surface potential can significantly influence attachment behavior during OA. Anisotropic electrostatic interactions, created by competing ions with facet-dependence, induce approaching NPs to preferentially orient before attachment. This surface potential evolves dynamically in response to changes in the relative surface coverages of the competing ions as surface area decreases during repeated OA events, potentially switching the direction of OA. Unlike intrinsic vdW and dipole–dipole interactions, electrostatic interactions can be finely and dynamically tuned across all surfaces by adjusting solution chemistry, such as pH, electrolyte type, and concentration. While electrostatic interactions have long been recognized as important in NP assembly, their role has typically been considered in terms of static attractive or repulsive forces rather than as dynamic, facet-specific torques capable of switching attachment pathways in real time. This suggests that the dynamic manipulation of these facet-specific electrostatic interactions and resultant torques provides an underexplored strategy for engineering NP assemblies, enabling precise spatial and temporal control over the OA process.”

9. Wang, Y. *et al.* Particle-based hematite crystallization is invariant to initial particle morphology. *Proceedings of the National Academy of Sciences* **119**, e2112679119 (2022).
29. Liu, L. *et al.* Effect of Solvent Composition on Non-DLVO Forces and Oriented Attachment of Zinc Oxide Nanoparticles. *ACS Nano* **18**, 16743–16751 (2024).
30. Zhang, X. *et al.* Direction-specific van der Waals attraction between rutile TiO_2 nanocrystals. *Science* **356**, 434–437 (2017).
34. Su, S. *et al.* Facet-Dependent Surface Charge and Hydration of Semiconducting Nanoparticles at Variable pH. *Advanced Materials* **33**, 2106229 (2021).
35. Liang, Y. *et al.* Facet-dependent surface charge and Pb^{2+} adsorption characteristics of hematite nanoparticles: CD-MUSIC-eSGC modeling. *Environmental Research* **196**, 110383 (2021).
36. Chatman, S., Zarzycki, P. & Rosso, K. M. Surface potentials of (001), (012), (113) hematite ($\alpha\text{-Fe}_2\text{O}_3$) crystal faces in aqueous solution. *Phys. Chem. Chem. Phys.* **15**, 13911 (2013).

- We also significantly restructured and rewrote the dense sections on "*The role of surface-potential in controlling NP alignment*" and "*The role of ion competition in driving variations in surface potential*" as follows. The revised versions now present the complex data in a more logical, top-down narrative to improve readability.

Revised text (Page 4, Line 38; in the main text):

“... After Pt reduction, the concentrations of all related ions stabilize at $[H^+]=4.75$, $[K^+]=2.9$, $[Cl^-]=5.8$, and $[HCOO^-]=1.85$ mM (Supplementary Note 4, Supplementary Table 1).

This indicates that the subsequent switch in attachment behavior, which occurs much later (at ~45 min), is not caused by ongoing changes in bulk solution chemistry. Instead, it must be driven by the evolution of the NP surface chemistry itself. As the ~3 nm NPs (**Fig. 3a**) attach and fuse into larger crystals (~120 nm, **Fig. 3e**), the total NP surface area in solution (A_{tot}) decreases by ~30 times (Supplementary Note 5). This drastic decrease in A_{tot} available for ion adsorption would alter the surface coverage of the various ions on the different facets over time, thereby modifying their surface potential.

To test this, we measured the zeta potential (ζ)—a proxy for surface potential—of surfaces dominated by {100} and {111} facets as a function of A_{tot} under conditions mimicking the stage ii–iii (Supplementary Figs. 14 and 15). The concentration of the initially generated NPs and A_{tot} are estimated to be 2.5 μ M and 30 m^2/L , respectively, considering the primary 3 nm NPs are truncated octahedra with {100} and {111} facets (Supplementary Fig. 16, Supplementary Note 5)²⁶. The concentrations of H^+ , K^+ , Cl^- , and $HCOO^-$ are reproduced by mixing KCl, HCl, and HCOOH (#2 in Supplementary Table 1).

The results showing that both {100} and {111} surfaces become more negatively charged with decreasing NP concentration (Supplementary Figs. 15 and 17) supports the hypothesis that the surfaces are undersaturated with respect to ion adsorption, making them highly sensitive to the change in A_{tot} ^{27,28}. This sensitivity leads to a striking anisotropic behavior between {100} and {111} surfaces (**Fig. 5c**). At a high A_{tot} (30 m^2/L , representing stage ii), the {100} surface exhibits a smaller ζ value (−0.7 mV) than the {111} surface (−3.2 mV). As A_{tot} decreases to 3 m^2/L (representing stage iii), ζ_{111} changes slightly from −3.2 mV to −4.5 mV, while ζ_{100} becomes more negative, dropping from near-neutral to −7.5 mV, which is then below ζ_{111} . This demonstrates a clear crossover in the relative electrostatic repulsion of the two surfaces during Pt mesocrystal growth via repeated OA events.”

Revised text (Page 6, Line 7; in the main text):

“To understand the crossover in ζ between the {100} and {111} surfaces (**Fig. 5c**), we investigated the competitive ion adsorption of the key ions in solution: H^+ , K^+ , Cl^- , and $HCOO^-$. Our experiments reveal that the adsorption behavior of these ions is highly sensitive to A_{tot} and that a complex competition between them dictates the surface potential of each facet.

First, we established the relative roles of the positive ions. Comparing HCl and KCl (**Fig. 6a**), both surfaces are less negatively charged in the presence of H^+ compared to K^+ , regardless of A_{tot} , indicating more H^+ is adsorbed than K^+ . When half of the K^+ is replaced with H^+ , this effect was more pronounced on the $\{100\}$ surface, particularly at high A_{tot} (≥ 10 m^2/L), where a positive $\Delta\zeta_H$ (the change in ζ upon replacing half of K^+ with H^+) is larger than on the $\{111\}$ surface (**Fig. 6b**). This difference indicates a higher affinity for H^+ on the $\{100\}$ surface, which dominates its surface charge in stage ii where $\{100\}$ OA occurs. In contrast, at low A_{tot} (3 m^2/L), H^+ has little effect on the change in ζ for either surface with $\Delta\zeta_H \sim 0$ (**Fig. 6b**).

Next, we investigated the influence of negative ions. In both HCl and KCl, the $\{100\}$ surface is more negatively charged than the $\{111\}$ surface, regardless of A_{tot} , indicating greater Cl^- adsorption on the $\{100\}$ surface (**Fig. 6a**). We then added various amounts of HCOOH in the solution containing 2.9 mM HCl and 2.9 mM KCl. Defining $\Delta\zeta_{HCOO}$ as the change in ζ upon adding HCOOH (Supplementary Fig. 15, **Fig. 6c**), at high A_{tot} (≥ 10 m^2/L), we find the addition of $HCOO^-$ has a minimal effect on the change in ζ for either surface regardless of [HCOOH]. However, at low A_{tot} (3 m^2/L), when [HCOO⁻] exceeds 1.14 mM, the same amount of $HCOO^-$ has a dramatic effect, making both surfaces less negatively charged with a $\Delta\zeta_{HCOO}$ of up to 14.3 mV. This suggests there is a complex competition between negatively charged Cl^- and $HCOO^-$ when available surface area is scarce.

To directly confirm this competitive adsorption, we used time-of-flight secondary ion mass spectrometry (TOF-SIMS) to measure the surface-bound ions at high and low A_{tot} (**Fig. 6d**, Supplementary Fig. 18). The analyses were conducted on the $\{100\}$ and $\{111\}$ surfaces with A_{tot} of 30 m^2/L and 3 m^2/L under the same conditions as in **Fig. 5c** to qualitatively analyze the relative changes in ion adsorption (Supplementary Note 6). As A_{tot} decreases, $HCOO^-$ adsorption increases on both surfaces by similar factors. In addition, H^+ and Cl^- adsorption increases on the $\{100\}$ surface, but decreases on the $\{111\}$ surface, demonstrating distinct ion competition on the two surfaces. While K^+ may also participate in competitive adsorption, here we have focused on H^+ because its adsorption dominates that of the positive ions (**Fig. 6a**)."

- In Fig. 1, we added a new schematic diagram that clearly illustrates the OA direction and the resulting morphology, aiding in immediate visual understanding.

Revised Fig. 1: Morphological analysis of branched cubic Pt mesocrystals.

a, Ex situ TEM image of synthesized mesocrystals formed from 1.45 mM of K_2PtCl_4 and 52 mM of HCOOH. Inset, schematic of a branched cubic mesocrystals showing the cube-shaped core (yellow) and branches (purple) growing from the cube faces. **b,c**, SAED pattern (**b**) and Scanning TEM image (**c**) from a single mesocrystal. Inset in **b**, corresponding TEM image. Scale bar, 40 nm. **d,e**, HR-TEM images (**d**) and corresponding FFT patterns (**e**) of cubic core obtained at 45 min, with an outline showing $\{100\}$ attachments between NPs. **f,g**, HR-TEM images (**f**) and corresponding FFT patterns (**g**) of branches showing they result from $\{111\}$ attachments between NPs to form $\{111\}$ -aligned nanorods. NPs are indicated by dotted circles and attachment plane is indicated by solid lines in **d,f**. Images in **a–c** are observed 2 days after the reaction. Z.A., zone axis. **h**, Schematics showing the OA direction and the resulting morphology.

- Similarly, we added a new schematic to Fig. 3 that visually represents the entire OA pathway including the switch in facet selection and the morphological evolution leading to the final branched cubic mesocrystal.

Revised Fig. 3: Evolution of branched cubic mesocrystals via OA of primary NPs.

a–e, Representative cryo-TEM images at different reaction times showing the multi-step crystallization of Pt mesocrystals going from NPs to clusters (stage i), to core/shell structures (stage ii), to branched cubes (stage iii). **f–j**, SAED patterns at the reaction time points; cluster at

15 min (**f**), core/shell structures with thick shell (**g**) and thin shell (**h**) at 27 min, cubes without branches at 43 min (**i**), and branched cubes at 66 min (**j**), respectively. Inset, corresponding cryo-TEM images of each single structure. Scale bars, 50 nm. **k–n**, HR-cryo-TEM images observed at 13 min and corresponding FFT patterns showing $\langle 100 \rangle$ alignment (**k,l**) and $\langle 111 \rangle$ alignment (**m,n**) between NPs within a cluster. **o–q**, HR-cryo-TEM image of a core/shell structure observed at 25 min (**o**) and corresponding FFT patterns of the core (**p**) and shell (**q**) marked by dashed lines in **o**. Z.A., zone axis. **r**, Schematic showing the evolution of branched cubic mesocrystals via OA involving a switch in the facet upon which attachment occurs.

We are confident that these substantial revisions directly address the concerns raised by the reviewer and the revised manuscript is now clear, accessible, and demonstrates the timeliness and importance of our work, making it suitable for publication in *Nature Communications*.

Reviewer 2

General comments: *This is a thoroughly researched work that utilizes an extensive battery of modern experimental techniques. The research has a great internal logic, despite the complexity of the problem it deals with. I like the article.*

The understanding of the morphogenesis of crystalline materials created by the assembly of nanoparticles through oriented aggregation - also referred to as mesocrystals by some research groups - is typically explained by forces between the particles, including electrostatic repulsion, which can be tuned by solution chemistry. The authors claim that anisotropies in surface potential, driven by competitive ion adsorption, can alter the selection of faces during the oriented aggregation of nanoparticles.

The authors perform experiments varying the ratio of reactants and adding either KCl or HCl, and follow the effect on morphology. The results are clear, and I agree with the explanation provided by the authors well supported by TEM and LPTEM videos.

Comment 1: *However, the main experiment of the authors, as written in the section “Synthesis and structural characterization of branched Pt cubes”, describes the generation of two consecutive types of oriented aggregation, namely the growth of a cubic core through {100} attachment and later at 40-45 minutes the extension of {111}-oriented branches out from the cubic core through {111} attachment. According to the experimental description, it does not appear that there is any change in the solution chemistry during the experiment beyond that caused by the formation of Pt particles, which is negligible. In fact, as shown in Figure 4a, neither the pH nor the UV absorbance changes significantly during the experiment and remain almost constant at 40-45 minutes, when the morphological transition is recorded. The hypothesis that competitive ion adsorption can explain the transition is, in my opinion, an ad hoc hypothesis. It could be accepted in the best of the cases for this particular experiment. To make the explanation universal, I strongly suggest that the authors compare the morphogenetic power of ion adsorption with that of solution composition and the solution chemistry independent dipole–dipole interactions and anisotropic van der Waals (vdW) interactions.*

Response 1: We appreciate the reviewer's careful examination of our experimental design and their valuable insights. The suggestion that we compare the morphogenetic power, i.e. anisotropic torque, from electrostatic interactions with that of other potential interactions to enhance the universality of our explanation is an excellent one.

In summary, each torque arising from dipole–dipole and van der Waals (vdW) interactions can be considered with contributions derived from both the shape effect and the intrinsic material properties. We note that the primary NPs are either spherical NPs (at early growth stage, Fig. 2) or a truncated octahedron (at later growth stage, Supplementary Fig. 16). Dipole–dipole interactions are insignificant because spherical or truncated octahedral Pt NPs are centrosymmetric, cancelling out any net dipole moment. This contrasts with materials like ZnO, which have inherent dipoles due to their crystal structure. vdW torques are minimal due to the spherical-like geometry of the Pt NPs and the isotropic dielectric properties of FCC metals like Pt, making the Hamaker constant effectively direction-independent. Because both dipole–dipole and

vdW torques are tied to intrinsic and largely constant material properties (shape, symmetry, dielectric function), they do not change dynamically during NP growth. This makes them unlikely causes for observed changes in attachment orientation during oriented attachment (OA). In contrast, electrostatic torques, which depend on ion adsorption and surface potentials that can change with conditions (e.g., pH), are variable and directional, making them the more dominant factor in such transitions.

Detailed explanations are as follows.

Dipole-dipole interaction: Although perfectly spherical and symmetric Pt NPs do not possess permanent dipole moments, structural or chemical asymmetries within the particles can result in nonuniform charge distributions that give rise to a net dipole moment. The truncated octahedron, however, is a highly symmetric structure, characterized in particular by inversion symmetry. In this geometry, each face has an identical counterpart on the opposite side of the particle, relative to its center. Since the dipole moment is a vector quantity that arises from asymmetric charge distributions, any dipole contributions associated with surface charges on individual facets are expected to be cancelled out due to this symmetry. Specifically, the dipole moments associated with each pair of square {100} facets and each pair of hexagonal {111} facets cancel one another. Even if the surface charge densities of the {111} and {100} facets differ, the overall distribution remains balanced because each type of facet is symmetrically arranged. As a result, the vector sum of dipole contributions from all hexagonal facets is 0, as is the sum from all square facets. Therefore, the net dipole moment of an ideal truncated octahedral Pt NP would be 0.

For the case of ZnO in contrast to truncated octahedral Pt, it possesses a permanent dipole moment due to its inherently asymmetrical internal crystal structure, whether spherical or symmetrical [*Nat. Commun.* **2020**, *11*, 1045]. This intrinsic polarity creates dipole-dipole interactions with distinct orientations between particles, so the anisotropic torque generated by these interactions can be a crucial factor to consider alongside torques by other interparticle interactions.

vdW interaction: Any nonspherical particle can experience a torque from vdW interactions due to its shape anisotropy [*J. Colloid Interface Sci.* **2023**, *652*, 1974; *ACS Nano* **2024**, *18*, 32386]. The truncated octahedron belongs to the family of Archimedean polyhedra, in which all vertices lie on a common circumscribing sphere. Its overall geometry closely approximates a sphere so that the geometric transition between the faces and vertices of the truncated octahedron is much less pronounced compared to a distinct example used in the aforementioned studies (i.e., cubic particles). This makes the variation in interaction energy for relative particle orientations insignificant. Thus, due to its high symmetry and near-sphericity, the vdW torque can be considered negligible when compared to that of more anisotropic shapes.

The Hamaker constant (A) is usually considered as a scalar property for a given material, derived from the frequency-dependent dielectric function of materials by Lifshitz theory [*Sov. Phys. JETP* **1956**, *2*, 73]. In Lifshitz theory extended to anisotropic crystals, materials have different dielectric constants along different principal axes, meaning that the Hamaker constant becomes orientation-dependent [*J. Adhesion* **1972**, *3*, 259]. However, in face-centered cubic (FCC) metals such as Pt, the dielectric function is highly isotropic due to the crystal's cubic symmetry. This isotropy implies

an isotropic or nearly isotropic dielectric response. In other words, the polarizability (α) does not vary significantly with crystallographic orientation (i.e., $\{100\}$, $\{111\}$) or with the angle (θ) between interacting surfaces (i.e., relative particle orientation). To illustrate this aspect further, one can utilize the Hamaker's seminal work [*Physica* **1937**, 4, 1058]; the Hamaker constant is proportional to the square of the polarizability. This can lead to a simple rationale that an angular dependence would be associated with a polarizability difference in directionality on the specific facet. Thus, the Hamaker constant is unlikely to be orientation- and angular-dependent, resulting in negligible differences in the angular derivative ($\partial A(\theta)/\partial \theta$), and consequently in the vdW torque.

Importantly, both dipole–dipole torque and vdW torque originate from intrinsic material properties and therefore remain constant over time during growth process even when something as basic as the total surface area is changing, as in the case of the Pt mesocrystals studied here. This implies that they are not directly responsible for the switch in planes of attachment. Instead, the dominant factor is more likely the electrostatic torque resulting from ion competition, along with its directional switching. This case—that electrostatic torque dominates over other interactions like vdW and dipole-dipole torques—is not specific to the Pt NPs studied here but can be generalized to a wide range of materials, given that: 1) inherently polar crystals like ZnO are an exception rather than the rule, 2) many nanocrystals, particularly those synthesized via common methods, adopt either spherical or other simple geometric shapes with high symmetry, which minimizes shape anisotropy, and 3) the polarizability of adjacent facets is similar for any high-symmetry crystal class, resulting in a negligible orientation-dependent vdW torque. In contrast, the surface potential of adjacent facets will generally differ in magnitude and, often, in sign over significant pH ranges. For example, in a non-polar oxide like Fe₂O₃ (hematite), $\{100\}$, $\{012\}$, and $\{113\}$ facets exhibit different pH dependence of surface potential due to differences in atomic arrangements and termination on facets [*Phys. Chem. Chem. Phys.* **2013**, 15, 13911]. Similarly, in the cubic perovskite semiconductor SrTiO₃, the $\{100\}$ facets can be terminated with either SrO or TiO₂ layers, being intrinsically electrically neutral, while the $\{110\}$ facets are polar with two highly charged possible terminations, SrTiO⁴⁺ and (2O)⁴⁻, resulting in facet-dependent charging behavior upon varying pH [*Adv. Mater.* **2021**, 33, 210622]. Therefore, our study that correlates facet-dependent electrostatic interactions to orientational motions of nanocrystals provides important physical insights on OA for a wide range of materials. Furthermore, if the interplay among the three types of torque can be leveraged to control assembly orientation involving materials with anisotropic Hamaker constants, permanent dipole moments, or anisotropic shapes, it can be extended to a strategy for material synthesis via directed assembly for virtually any crystal class.

This last statement may seem to be inapplicable to cubic systems or any system that is characterized by growth habits for which the faces are crystallographically identical. However, that is not the case, because all crystallographic directions are represented nonetheless, even if only at rounded corners. Recent results on growth of hematite by OA show that, at a single value of the pH, crystals with very different growth habits all attach along the $\langle 001 \rangle$ direction *even when there is no (001) facet*, as shown in the adjacent figure (Wang et al. 2022, *Proc. Nat'l Acad. Sci.* **119**, e2112679119).

- Considering the above discussion, we took the reviewer's advice and modified the main text to make the generality of the results clearer.

Revised text (Page 5, Line 26; in the main text):

“... This suggests that the anisotropy in the electrostatic interactions leads to a torque that pre-aligns the particles prior to contact.

Although such torques have been observed in other systems as a consequence of both dipole–dipole interactions^{10,29} and anisotropic van der Waals (vdW) interactions^{30,31}, we note that these sources of torque are unlikely to be the cause in our system (Supplementary Note 8). First, individual Pt particles are nearly spherical (truncated octahedra, Supplementary Fig. 16) and possess a symmetric face-centered cubic (FCC) structure. Given this high degree of symmetry, significant torques originating from intrinsic material properties (e.g., dipole moment, anisotropic dielectric response) are not expected. Furthermore, any such intrinsic torques would remain constant during the growth process and therefore cannot account for the dynamic switching of the attaching surfaces.”

Revised text in the discussion (Page 7, Line 43; in the main text):

“The solution to this puzzle lies in physical torques that rotate and align the NPs against inherent Brownian torques. While torques created by inherent interparticle interactions, such as vdW and dipole–dipole, are known to contribute to this process (Supplementary Note 8)^{10,29–31}, the findings presented here now extend this concept, identifying anisotropic electrostatic interactions as a new, powerful, and externally tunable source of such torques (unlike vdW and dipole–dipole torques, which originate from the intrinsic properties of the materials). Moreover, mechanism of using competitive ion adsorption to generate tunable electrostatic torques is not limited to the Pt system but represents a generalizable strategy for rationally designing complex nanomaterials, particularly when pH is recognized as another means for manipulating surface potentials through H⁺ and OH[−] adsorption, because this strategy relies on two ubiquitous features of crystalline materials: the existence of distinct facets and the facet-dependent nature of their surface chemistry. Thus, this approach can be readily extended to other noble metals, oxides, and semiconductors, where ion- or pH-sensitive charge of surface terminations provides a natural handle for electrostatic manipulation^{19,34–36}. While this approach may seem to be inapplicable to cubic systems or any system that is characterized by growth habits for which the faces are crystallographically identical, that is not the case, because all crystallographic directions are represented nonetheless, even if only at rounded corners. Recent results on growth of hematite (Fe₂O₃) by OA show that crystals with very different growth habits all attach along the <001> direction even when there is no (001) facet present and thus attachment must occur on the corners of the nanocrystals⁹.”

9. Wang, Y. *et al.* Particle-based hematite crystallization is invariant to initial particle morphology. *Proceedings of the National Academy of Sciences* **119**, e2112679119 (2022).

29. Liu, L. *et al.* Effect of Solvent Composition on Non-DLVO Forces and Oriented Attachment of Zinc Oxide Nanoparticles. *ACS Nano* **18**, 16743–16751 (2024).
30. Zhang, X. *et al.* Direction-specific van der Waals attraction between rutile TiO₂ nanocrystals. *Science* **356**, 434–437 (2017).
34. Su, S. *et al.* Facet-Dependent Surface Charge and Hydration of Semiconducting Nanoparticles at Variable pH. *Advanced Materials* **33**, 2106229 (2021).
35. Liang, Y. *et al.* Facet-dependent surface charge and Pb²⁺ adsorption characteristics of hematite nanoparticles: CD-MUSIC-eSGC modeling. *Environmental Research* **196**, 110383 (2021).
36. Chatman, S., Zarzycki, P. & Rosso, K. M. Surface potentials of (001), (012), (113) hematite (α -Fe₂O₃) crystal faces in aqueous solution. *Phys. Chem. Chem. Phys.* **15**, 13911 (2013).

- We added additional information in Supplementary Note 8 as follows.

“Supplementary Note 8. Torque arising from dipole–dipole and van der Waals interactions

Each torque arising from dipole–dipole and van der Waals (vdW) interactions can be considered with contributions derived from the shape effect and the intrinsic material properties, respectively. We note that the primary NPs are a truncated octahedron (Supplementary Fig. 16).

Dipole-dipole interaction: Although perfectly spherical and symmetric Pt NPs do not possess permanent dipole moments, structural or chemical asymmetries within the particles can result in nonuniform charge distributions that give rise to a net dipole moment. The truncated octahedron, however, is a highly symmetric structure, characterized in particular by inversion symmetry. In this geometry, each face has an identical counterpart on the opposite side of the particle, relative to its center. Since the dipole moment is a vector quantity that arises from asymmetric charge distributions, any dipole contributions generated by surface charges on individual facets cancel out due to this symmetry. Specifically, the dipole moments associated with each pair of square {100} facets and each pair of hexagonal {111} facets cancel one another. Even if the surface charge densities of the {111} and {100} facets differ, the overall distribution remains balanced because each type of facet is symmetrically arranged. As a result, the vector sum of dipole contributions from all hexagonal facets is 0, as is the sum from all square facets. Therefore, the net dipole moment of an ideal truncated octahedral Pt NP is 0.

Some materials, such as ZnO, have a permanent dipole moment due to their inherently asymmetrical internal crystal structure, whether spherical or symmetrical¹⁰. This intrinsic polarity creates dipole-dipole interactions with distinct orientations between particles, so the anisotropic torque generated by these interactions can be a crucial factor to consider alongside torques by other interparticle interactions.

vdW interaction: Any nonspherical particle can experience a torque from vdW interactions due to its shape anisotropy^{11,12}. The truncated octahedron belongs to the family of Archimedean polyhedra, in which all vertices lie on a common circumscribing sphere. Its overall geometry closely approximates a sphere so that the geometric transition between the faces and vertices of the truncated octahedron is much less pronounced compared to a distinct example such as cubic particles. This makes the variation in interaction energy for relative particle orientations insignificant. Thus, due to its high symmetry and near-sphericity, the vdW torque can be considered negligible when compared to that of more anisotropic shapes.

The Hamaker constant (A) is usually considered as a scalar property for a given material, derived from the frequency-dependent dielectric function of materials by Lifshitz theory¹³. In Lifshitz theory extended to anisotropic crystals, materials have different dielectric constants along different principal axes, meaning that the Hamaker constant becomes orientation-dependent¹⁴. However, in face-centered cubic (FCC) metals such as Pt, the dielectric response is highly isotropic due to the crystal's cubic symmetry. This isotropy implies an isotropic or nearly isotropic dielectric response. In other words, the polarizability (α) does not vary significantly with crystallographic orientation (i.e., $\{100\}$, $\{111\}$) or with the angle (θ) between interacting surfaces (i.e., relative particle orientation). To illustrate this aspect further, one can utilize the Hamaker's seminal work¹⁵; the Hamaker constant is proportional to the square of the polarizability. This can lead to a simple rationale that an angular dependence would be associated with a polarizability difference in directionality on the specific facet. Thus, the Hamaker constant is unlikely to be orientation- and angular-dependent, resulting in negligible differences in the angular derivative ($\partial A(\theta)/\partial \theta$), and consequently in the vdW torque.

Importantly, both dipole–dipole torque and vdW torque originate from intrinsic material properties and therefore remain constant over time during growth process of branched cubic mesocrystals. This implies that they are not directly responsible for the switch in planes of attachment. Instead, the dominant factor is more likely the electrostatic torque resulting from ion competition, along with its directional switching. This case—that electrostatic torque dominates over other interactions like vdW and dipole-dipole torques—is not specific to the Pt NPs studied here but can be generalized to a wide range of materials, given that: 1) inherently polar crystals like ZnO are an exception rather than the rule, 2) many nanocrystals, particularly those synthesized via common methods, adopt either spherical or other simple geometric shapes with high symmetry, which minimizes shape anisotropy, and 3) the polarizability of adjacent facets is similar for any high-symmetry crystal class, resulting in a negligible orientation-dependent vdW torque. In contrast, the surface potential of adjacent facets will generally differ in magnitude and, often, in sign, over significant pH ranges. For example, in a non-polar oxide like Fe₂O₃ (hematite), $\{100\}$, $\{012\}$, and $\{113\}$ facets exhibit different pH dependence of surface potential due to differences in atomic arrangements and termination on facets¹⁶. Similarly, in the cubic perovskite semiconductor SrTiO₃, the $\{100\}$ facets can be terminated with either SrO or TiO₂ layers, being intrinsically electrically neutral, while the $\{110\}$ facets are polar with two highly charged possible terminations, SrTiO⁴⁺ and (2O)⁴⁻, resulting in facet-dependent charging behavior upon

varying pH¹⁷. Therefore, our study that correlates facet-dependent electrostatic interactions to orientational motions of nanocrystals provides important physical insights on OA for a wide range of materials. Furthermore, if the interplay among the three types of torque can be leveraged to control assembly orientation involving materials with anisotropic Hamaker constants, permanent dipole moments, or anisotropic shapes, it can be extended to a strategy for material synthesis via directed assembly for virtually any crystal class.

This last statement may seem to be inapplicable to cubic systems or any system that is characterized by growth habits for which the faces are crystallographically identical. However, that is not the case, because all crystallographic directions are represented nonetheless, even if only at rounded corners. Recent results on growth of hematite by OA show that, at a single value of the pH, crystals with very different growth habits all attach along the <001> direction *even when there is no (001) facet*¹⁸.”

10. Liu, L. *et al.* Connecting energetics to dynamics in particle growth by oriented attachment using real-time observations. *Nat Commun* **11**, 1045 (2020).
11. Lee, J. *et al.* Effects of particle shape and surface roughness on van der Waals interactions and coupling to dynamics in nanocrystals. *Journal of Colloid and Interface Science* **652**, 1974–1983 (2023).
12. Lee, J. *et al.* Defect Self-Elimination in Nanocube Superlattices Through the Interplay of Brownian, van der Waals, and Ligand-Based Forces and Torques. *ACS Nano* **18**, 32386–32400 (2024).
13. Lifshitz, E. M. & Hamermesh, M. The theory of molecular attractive forces between solids. in *Perspectives in Theoretical Physics* (ed. Pitaevski, L. P.) 329–349 (Pergamon, Amsterdam, 1992). doi:10.1016/B978-0-08-036364-6.50031-4.
14. Parsegian, V. A. & Weiss, G. H. Dielectric Anisotropy and the van der Waals Interaction between Bulk Media. *The Journal of Adhesion* **3**, 259–267 (1972).
15. Hamaker, H. C. The London—van der Waals attraction between spherical particles. *Physica* **4**, 1058–1072 (1937).
16. Chatman, S., Zarzycki, P. & Rosso, K. M. Surface potentials of (001), (012), (113) hematite (α -Fe₂O₃) crystal faces in aqueous solution. *Phys. Chem. Chem. Phys.* **15**, 13911 (2013).
17. Su, S. *et al.* Facet-Dependent Surface Charge and Hydration of Semiconducting Nanoparticles at Variable pH. *Advanced Materials* **33**, 2106229 (2021).
18. Wang, Y. *et al.* Particle-based hematite crystallization is invariant to initial particle morphology. *Proceedings of the National Academy of Sciences* **119**, e2112679119 (2022).

Comment 2: *The videos are very informative. I suggest adding a scale bar to the videos, as it is currently only displayed at the caption of the videos, which is not very practical. Also, use real-*

time instead of 4x. Currently, they last only a few seconds, making it difficult to appreciate the information they provide fully.

Response 2: We are delighted to hear that the reviewer found our videos informative. We sincerely appreciate the constructive suggestions for their improvement, as enhancing their practical utility and clarity for the reader is very important to us. We added a scale bar directly to the videos and adjusted the video playback to real-time (1x speed).

Comment 3: *As I have already said, the work is of a great technical quality. However, the authors should make an effort to explain the importance and novelty of their work, i.e. the importance of competitive ion adsorption in controlling OA beyond this case of platinum nanoparticles aggregation.*

Response 3: Thank you for pointing out the need to articulate the broader importance of our findings better. To address this, we explicitly argued that the principle of using competitive ion adsorption to create electrostatic torques is a generalizable strategy and provided specific examples of how this mechanism can be extended to other important material systems, including other noble metals, metal oxides, and semiconductors, thereby highlighting the timeliness and potential impact of our work. We also added a discussion of the other interactions mentioned in Comment 1.

- We revised the introduction to better place this work in the context of key challenges and knowledge gaps in the field and the discussion to include the broader implications and generality of the mechanism.

Revised text in the introduction (Page 2, Line 2; in the main text):

“When crystalline materials form through assembly of nanoparticles (NPs), their properties are strongly influenced by the assembled architecture because phenomena like photon and electron scattering, electron–hole recombination, and dislocation generation depend on the material’s characteristic length scales and topology^{1–3}. **Oriented attachment (OA) has emerged as a key pathway for creating single-crystal-like structures with diverse morphologies^{4–7}.**

During OA, neighboring particles align and fuse along matching crystallographic planes, but intriguingly, they often do so with high facet-selectivity—certain crystallographic facets are repeatedly favored over others^{4,8,9}. This consistent selectivity is surprising because particles in solution constantly undergo random Brownian motion, leading to sampling of all possible orientations. Why particles consistently select specific crystallographic facets when attachment on any set of matched lattice planes would reduce the systems energy remains an unresolved question in the field. Answering that question would address one of the key challenges in designing crystal structures: understanding how to control and select specific facets for OA, thus enabling the tailoring of OA-based crystal growth and assembly.

Recent efforts have begun to unravel the complex interplay of interparticle forces that drive OA. For example, real-time imaging of OA in the ZnO system has established a relationship

between particle structure, interaction forces arising from ion-solvent correlations and dipolar interactions, and the resulting assembly dynamics¹⁰. Other studies in metal oxide systems have explored the influence of ion-correlation forces on OA¹¹. These studies show that OA kinetics and pathways are intricately tied to interfacial chemistry, which is governed by environmental factors, including electrolyte type and concentration, surface adsorbates, and pH^{12–17}. However, despite this progress, current understanding remains largely focused on interaction energetics, with little attention paid to how directional rotational alignment is achieved. Since facet-selective OA requires precise orientation at the moment of collision, this gap points to the need for a mechanism capable of generating directional torque, especially at nanometer distance.”

8. Pacholski, C., Kornowski, A. & Weller, H. Self-Assembly of ZnO: From Nanodots to Nanorods. *Angewandte Chemie International Edition* **41**, 1188–1191 (2002).
9. Wang, Y. *et al.* Particle-based hematite crystallization is invariant to initial particle morphology. *Proceedings of the National Academy of Sciences* **119**, e2112679119 (2022).
10. Liu, L. *et al.* Connecting energetics to dynamics in particle growth by oriented attachment using real-time observations. *Nat Commun* **11**, 1045 (2020).
11. Sushko, M. L. & Rosso, K. M. The origin of facet selectivity and alignment in anatase TiO₂ nanoparticles in electrolyte solutions: implications for oriented attachment in metal oxides. *Nanoscale* **8**, 19714–19725 (2016).
17. Jin, B. *et al.* Peptoid-Directed Formation of Five-Fold Twinned Au Nanostars through Particle Attachment and Facet Stabilization. *Angew Chem Int Ed* **61**, e202201980 (2022).

Revised text in the discussion (Page 7, Line 30; in the main text):

“OA has important implications for achieving novel material properties, because it allows NPs to spontaneously organize, enabling control over size and morphology that cannot be achieved through ion-by-ion crystal growth. The driving force behind OA is the net reduction in free energy that comes about with the reduction in surface area achieved when particles merge. However, minimization of free energy alone cannot account for the facet-selective nature of attachment, as interacting NPs cannot inherently distinguish between surfaces with higher or lower energy, and attachment on any matched lattice plane will reduce the free energy, even if attachment on one specific set of planes reduces it the most. Thus, when particles interact with random orientations driven by Brownian motions, facet-selectivity is only possible if either the barriers to attachment on all other planes are too large to overcome, or attachment is a reversible process, allowing the particles to sample all possible configurations. Even then, there would be a distribution of attachment planes that reflect the relative changes in free energy. Neither of these situations is present in the Pt NP system.

The solution to this puzzle lies in physical torques that rotate and align the NPs against inherent Brownian torques. While torques created by inherent interparticle interactions,

such as vdW and dipole–dipole, are known to contribute to this process (Supplementary Note 8)^{10,29–31}, the findings presented here now extend this concept, identifying anisotropic electrostatic interactions as a new, powerful, and externally tunable source of such torques (unlike vdW and dipole–dipole torques, which originate from the intrinsic properties of the materials). Moreover, mechanism of using competitive ion adsorption to generate tunable electrostatic torques is not limited to the Pt system but represents a generalizable strategy for rationally designing complex nanomaterials, particularly when pH is recognized as another means for manipulating surface potentials through H⁺ and OH⁻ adsorption, because this strategy relies on two ubiquitous features of crystalline materials: the existence of distinct facets and the facet-dependent nature of their surface chemistry. Thus, this approach can be readily extended to other noble metals, oxides, and semiconductors, where ion- or pH-sensitive charge of surface terminations provides a natural handle for electrostatic manipulation^{19,34–36}. While this approach may seem to be inapplicable to cubic systems or any system that is characterized by growth habits for which the faces are crystallographically identical, that is not the case, because all crystallographic directions are represented nonetheless, even if only at rounded corners. Recent results on growth of hematite (Fe₂O₃) by OA show that crystals with very different growth habits all attach along the <001> direction even when there is no (001) facet present and thus attachment must occur on the corners of the nanocrystals⁹

The collective findings in this study demonstrate that the interplay of ion adsorption and resultant surface potential can significantly influence attachment behavior during OA. Anisotropic electrostatic interactions, created by competing ions with facet-dependence, induce approaching NPs to preferentially orient before attachment. This surface potential evolves dynamically in response to changes in the relative surface coverages of the competing ions as surface area decreases during repeated OA events, potentially switching the direction of OA. Unlike intrinsic vdW and dipole–dipole interactions, electrostatic interactions can be finely and dynamically tuned across all surfaces by adjusting solution chemistry, such as pH, electrolyte type, and concentration. While electrostatic interactions have long been recognized as important in NP assembly, their role has typically been considered in terms of static attractive or repulsive forces rather than as dynamic, facet-specific torques capable of switching attachment pathways in real time. This suggests that the dynamic manipulation of these facet-specific electrostatic interactions and resultant torques provides an underexplored strategy for engineering NP assemblies, enabling precise spatial and temporal control over the OA process.”

9. Wang, Y. *et al.* Particle-based hematite crystallization is invariant to initial particle morphology. *Proceedings of the National Academy of Sciences* **119**, e2112679119 (2022).
29. Liu, L. *et al.* Effect of Solvent Composition on Non-DLVO Forces and Oriented Attachment of Zinc Oxide Nanoparticles. *ACS Nano* **18**, 16743–16751 (2024).
30. Zhang, X. *et al.* Direction-specific van der Waals attraction between rutile TiO₂ nanocrystals. *Science* **356**, 434–437 (2017).

34. Su, S. *et al.* Facet-Dependent Surface Charge and Hydration of Semiconducting Nanoparticles at Variable pH. *Advanced Materials* **33**, 2106229 (2021).
35. Liang, Y. *et al.* Facet-dependent surface charge and Pb²⁺ adsorption characteristics of hematite nanoparticles: CD-MUSIC-eSGC modeling. *Environmental Research* **196**, 110383 (2021).
36. Chatman, S., Zarzycki, P. & Rosso, K. M. Surface potentials of (001), (012), (113) hematite (α -Fe₂O₃) crystal faces in aqueous solution. *Phys. Chem. Chem. Phys.* **15**, 13911 (2013).

Reviewer 3

General comments: *In this manuscript the authors investigate how ion adsorption can induce time-dependent, facet-specific changes in surface potential that govern the oriented attachment and morphology of Pt mesocrystals. They combine direct observations from LP-EM and cryo-TEM, with mechanistic interpretation following zeta potential measurements, TOF-SIMS, and DFT calculations. The authors show branched Pt NP grow through two stages of oriented attachment which can be manipulated through dynamic, facet-specific changes in surface potential via adsorption of Cl^- , H^+ , and $HCOO^-$ ions. The authors claim this strategy can be generally applied to the synthesis of complex nanostructures.*

The study is thorough; the methodology is sound and offers a detailed mechanistic understanding of forces governing Pt NP assemblies.

A few comments that should be addressed prior to publication:

Comment 1: *The landscape of previous works on material synthesis through OA is not detailed enough in the introduction, making it difficult to grasp the novelty of the current work easily. Specific examples of findings from previous studies like Liu, Nat Commun 11, 1045 (2020) (where the relationships between structure, forces and response dynamics in OA of ZnO NP was established) could help.*

The following statement is too general “However, manipulating the OA process to produce novel architectures at will, based on a fundamental understanding of the underlying interparticle forces has not been possible.” Examples of previous works: (1) Biao Jin, Angew Chemie 16, e202201980 (2022), (2) Sushko, Nanoscale 8, 19714-19725 (2016)

Response 1: We appreciate the reviewer's insightful feedback. We revised the introduction to better contextualize our work within the field. We incorporated the findings from Liu et al. (2020) to highlight the significant progress being made in fundamentally understanding the relationship between interparticle forces and OA dynamics. As suggested, we also included the previous work by Jin et al. (2022) and Sushko et al. (2016). Based on this more detailed background, we replaced the overly general statement. The revised text now clarifies that mechanisms and dynamic control to achieve facet-specificity remains a major challenge.

- We revised the introduction as below.

Revised text in the introduction (Page 2, Line 2; in the main text):

“When crystalline materials form through assembly of nanoparticles (NPs), their properties are strongly influenced by the assembled architecture because phenomena like photon and electron scattering, electron–hole recombination, and dislocation generation depend on the material’s characteristic length scales and topology^{1–3}. **Oriented attachment (OA) has emerged as a key pathway for creating single-crystal-like structures with diverse morphologies^{4–7}.**

During OA, neighboring particles align and fuse along matching crystallographic planes, but intriguingly, they often do so with high facet-selectivity—certain crystallographic facets are

repeatedly favored over others^{4,8,9}. This consistent selectivity is surprising because particles in solution constantly undergo random Brownian motion, leading to sampling of all possible orientations. Why particles consistently select specific crystallographic facets when attachment on any set of matched lattice planes would reduce the systems energy remains an unresolved question in the field. Answering that question would address one of the key challenges in designing crystal structures: understanding how to control and select specific facets for OA, thus enabling the tailoring of OA-based crystal growth and assembly.

Recent efforts have begun to unravel the complex interplay of interparticle forces that drive OA. For example, real-time imaging of OA in the ZnO system has established a relationship between particle structure, interaction forces arising from ion-solvent correlations and dipolar interactions, and the resulting assembly dynamics¹⁰. Other studies in metal oxide systems have explored the influence of ion-correlation forces on OA¹¹. These studies show that OA kinetics and pathways are intricately tied to interfacial chemistry, which is governed by environmental factors, including electrolyte type and concentration, surface adsorbates, and pH^{12–17}. However, despite this progress, current understanding remains largely focused on interaction energetics, with little attention paid to how directional rotational alignment is achieved. Since facet-selective OA requires precise orientation at the moment of collision, this gap points to the need for a mechanism capable of generating directional torque, especially at nanometer distance.

Amongst the forces defining interparticle potentials, the repulsive electrostatic force is most strongly dependent on the chemistry of the NP surface as it relies on the surface potential, which can vary independently on different crystal facets^{18,19}. Here we show that the facet-specificity of attachment by Pt NPs can be manipulated through changes in the surface potential of distinct facets to drive a transition from attachment on Pt{100} to Pt{111}. This transition leads to a switch from growth of a cubic core through {100} attachment to the extension of {111}-oriented branches out from the cubic core through {111} attachment. Moreover, we find that the disparity in the surface potential between the two facets creates an electrostatic torque that is critical for ensuring facet specificity. Finally, we demonstrate that competitive ion adsorption underlies the changes in surface potential that leads to the transition from {100} to {111} attachment through torque-driven facet selection.”

8. Pacholski, C., Kornowski, A. & Weller, H. Self-Assembly of ZnO: From Nanodots to Nanorods. *Angewandte Chemie International Edition* **41**, 1188–1191 (2002).
9. Wang, Y. *et al.* Particle-based hematite crystallization is invariant to initial particle morphology. *Proceedings of the National Academy of Sciences* **119**, e2112679119 (2022).
10. Liu, L. *et al.* Connecting energetics to dynamics in particle growth by oriented attachment using real-time observations. *Nat Commun* **11**, 1045 (2020).
11. Sushko, M. L. & Rosso, K. M. The origin of facet selectivity and alignment in anatase TiO₂ nanoparticles in electrolyte solutions: implications for oriented attachment in metal oxides. *Nanoscale* **8**, 19714–19725 (2016).

17. Jin, B. *et al.* Peptoid-Directed Formation of Five-Fold Twinned Au Nanostars through Particle Attachment and Facet Stabilization. *Angew Chem Int Ed* **61**, e202201980 (2022).

Comment 2: *Could the switch between the growth of cubic core {100} and branching extensions {111} be visualized by LP-EM? Could the authors explain their rationale for showing the later stages of mesocrystal formation by cryo-TEM and not LP-EM? This should be addressed in the text.*

Response 2: We thank the reviewer for this important comment. While cluster and core/shell-like structures were successfully observed in our LPTEM experiments, the system did not progress to the final branched architecture seen in the fully developed mesocrystals. We believe this limitation arises for two reasons, both associated with the confined nature of the liquid-cell environment. First, the limited volume of the liquid cell as compared to the containers used to prepare samples for either ex situ or cryoTEM translates to a smaller number of primary nanoparticles. There simply may not be enough particles to reach the branching stage. Second, the cube-shaped cores already span an appreciable fraction of the liquid cell thickness, which is 100–200 nm. Thus, by the time the core has been generated, there is greatly restricted Brownian motion of NPs, due to interactions with the SiN_x window, and the increased solution viscosity within the confined liquid (typically 100–200 nm thick) likely suppressed the interparticle dynamics required for branch formation, especially in the region between at the periphery of ~100 nm-sized clusters and the SiN_x window [Langmuir **2015**, 31, 6956; Sci. Adv. **2021**, 7, abi5419]. In contrast, the cryo-TEM images were obtained from the bulk synthesis solution and were able to capture intermediate states at later stages of assembly. Although the resolution is low, the cryo-TEM data (Fig. R1, Supplementary Fig. S12b) suggests that NPs align in linear arrays pointing toward the solution, transitioning to branch growth.

Fig. R1. Magnified cryoTEM images at 15 min in Supplementary Fig. S12b showing NPs aligning in linear arrays pointing toward the solution and the onset of branching.

- We clarified this point in the revised manuscript to better explain our rationale for using cryo-TEM to visualize the later stages of growth.

Revised text (Page 3, Line 31; in the main text):

“... These LPTEM results confirm that the NPs form rapidly and serve as building blocks in early-stage growth, first associating and then rearranging before OA begins (Fig. 2e). While

the LPTEM captured the early stage of growth, the system did not evolve into the final branched cubes. This limitation is likely due to the confined liquid-cell environment, either because the volume is too small for an adequate number of NPs to be generated for the particles to reach the branching stage and/or because restricted Brownian motion—caused by interactions with the SiN_x window—and increased viscosity in the ~100–200 nm-thick liquid layer suppressed the interparticle dynamics required for further growth, particularly near the periphery of ~100 nm-sized NP clusters²⁴.”

24. Verch, A., Pfaff, M. & de Jonge, N. Exceptionally Slow Movement of Gold Nanoparticles at a Solid/Liquid Interface Investigated by Scanning Transmission Electron Microscopy. *Langmuir* 31, 6956–6964 (2015).

Comment 3: *I find Supplementary Fig. 11 to be very helpful in visualizing the role of the different species on the morphology. The authors should consider moving it from the SI to the main text, perhaps as part of Fig. 4.*

Response 3: We appreciate the reviewer’s positive assessment of Supplementary Fig. 11. We have followed the suggestion to move it to the main text as Fig. 4. As a result of this modification, the original Figs. 4 and 5 have been renumbered along with and original Supplementary Figs. 12 to 24. The manuscript has been updated to reflect these changes.

Revised Fig. 4. Effect of ionic additives and precursor concentrations on the shape of branched cubic Pt mesocrystals.

a,b, TEM images showing the effect of adding KCl or HCl on the resulting shape. The concentration of KCl added is 0, 2.9, and 5.8 mM, respectively in **a**, and the concentration of HCl added is 2.9 mM in **b**. The concentrations of K_2PtCl_4 and $HCOOH$ are 1.45 mM and 52 mM, respectively. **c**, TEM images of branched cubic Pt mesocrystals as a function of initial precursor concentrations showing the formation of longer branches with increasing $HCOO^-/Pt$ ratio. The synthesis conditions for each numbered image are listed in Supplementary Table 1. Insets, the corresponding SAED patterns of each TEM image. Scale bars, 5 nm^{-1} . All TEM images are observed 2 days after the reaction.

Comment 4: *In the discussion the authors state “the action of electrostatic torques must be a general phenomenon and may govern the facet selection of OA in a wide range of NP systems”, could the authors explain more specifically how this can be done? As switching mechanism relies on specific ion adsorption behaviours on Pt facets, which may not extend to other materials. Also, the statement “This suggests that the ability to manipulate electrostatic interactions and resultant*

electrostatic torques provides an unexplored strategy for engineering NP assemblies with spatial and temporal control of the OA process.” seems overstated. While specifically using ion-modulated, facet-specific electrostatic torques to trigger dynamic facet switching in OA is novel, the broader concept of spatial and temporal control of nanoparticle assembly through electrostatic manipulation is not.

Response 4: We thank the reviewer for critically assessing our claims. We agree that the specific switching mechanism observed in our system is strongly influenced by the competitive ion adsorption behavior of Pt facets and that this particular mechanism for manipulating facet selection is going to be limited to systems for which facet specific adsorption is manifest at a sufficient level and with trends that lead to a switch in the relative values of the surface potential. Having said that, many systems will undoubtedly exhibit such behavior. Given that it is true for Pt, one can anticipate that it will be so for other noble metals. More importantly, this behavior can be expected to occur in many oxides, because the distributions of cation, oxygen and hydroxyl

sites vary greatly between facets. However, our claim about the broad applicability of exploiting electrostatic torque arising from tunable facet-dependent surface potentials as a mechanism for manipulating OA explicitly referred to pH as a means for executing this strategy. That would provide a truly general approach, because the overwhelming majority of inorganic crystals have facet dependent points of zero charge. Take the AlOOH (boehmite) system as an example; the nanocrystals can be grown to express multiple facets, but due to the differences in the distribution of oxygen sites and the existence of sites with different coordination environments and, thus, different pK_a values, the points of zero charge are different for all potential facets, as shown in the figure above [L. Liu et al., 2023, *ACS Nano* 17, 15556]. Indeed, the facets of attachment observed in the boehmite system are pH dependent, but the thought of examining the behavior from the point of view of electrostatic torque has never been considered. Of course, the effect of pH change is also competitive ion adsorption; the ions happen to be H^+ and OH^- . Other systems for which we know that pH-dependent surface potentials have been measured include titania and hematite. Moreover, while facet specific surface potentials in the calcium carbonate system have not been measured, in both the aragonite and calcite phases the (001) planes are carbonate terminated while the common (104) facets of calcite and the (110) faces of aragonite have equal number densities of calcium and carbonate ions. Thus, the surface potentials and points of zero charge must differ. If pH dependent surface potentials are manifest in noble metals, oxides, and carbonates, then we can reasonably expect that they are manifest across a wide swath of crystal systems. Consequently, we believe we are on solid ground in suggesting that modulation of ionic composition or pH can lead to anisotropic surface potentials, thereby giving rise to electrostatic torques that influence the rotational alignment of NPs during OA for many systems other than Pt.

We also acknowledge the reviewer’s point regarding prior studies on electrostatic control in NP assembly. While electrostatic interactions as indeed a well-established factor in determining

pH dependence of surface potential in the boehmite system from Liu et al, 2023, *ACS Nano*, 17 15556.

interparticle interactions, their role has largely been considered in terms of relative magnitudes for attractive and repulsive forces, and not in terms of directional torques that promote rotational alignment between NPs. Our work demonstrates that facet-specific, ion-regulated electrostatic interactions can generate torques that align NPs along specific crystallographic orientations, and more importantly, that these torques can be dynamically changed over time, leading to temporal switching of the attaching facets. This facet-switching behavior is not a static equilibrium feature but one that can be varied in time to redirect the assembly pathway, either by relying on the differences in ion adsorption that naturally arise with decrease in total surface area, as is the case in the current work, or through intentional time-dependent manipulation as is easily achieved through changes to ion concentration or the pH of a solution. Indeed, one can even actuate such a change nearly instantaneously with photoacids and photobases, which is a strategy that one of us has used to repeatedly drive oriented assembly and disassembly of proteins through protonation and deprotonation of surface sites using UV illumination [H. Shen et al. *Nature Nanotech.* 2024, **19**, 1016]. Thus, the concept of intentionally switching OA between specific facets deterministically over time is not a wild idea. In our opinion, it is truly achievable and will be the subject of our follow-on research. In summary, to our knowledge, this phenomenon of switching the progression of OA-based crystal growth between distinct facets of attachment through the use of electrostatic torque has not been previously reported. Furthermore, we believe that this approach enables control over assembly dynamics by tuning the solution chemistry in real time during synthesis, offering a new strategy for precise spatiotemporal control over OA processes through the manipulation of surface potentials.

- We revised the text to better distinguish our contribution.

Revised text in the discussion (Page 7, Line 30; in the main text):

“OA has important implications for achieving novel material properties, because it allows NPs to spontaneously organize, enabling control over size and morphology that cannot be achieved through ion-by-ion crystal growth. The driving force behind OA is the net reduction in free energy that comes about with the reduction in surface area achieved when particles merge. However, minimization of free energy alone cannot account for the facet-selective nature of attachment, as interacting NPs cannot inherently distinguish between surfaces with higher or lower energy, and attachment on any matched lattice plane will reduce the free energy, even if attachment on one specific set of planes reduces it the most. Thus, when particles interact with random orientations driven by Brownian motions, facet-selectivity is only possible if either the barriers to attachment on all other planes are too large to overcome, or attachment is a reversible process, allowing the particles to sample all possible configurations. Even then, there would be a distribution of attachment planes that reflect the relative changes in free energy. Neither of these situations is present in the Pt NP system.

The solution to this puzzle lies in physical torques that rotate and align the NPs against inherent Brownian torques. While torques created by inherent interparticle interactions, such as vdW and dipole–dipole, are known to contribute to this process (Supplementary Note 8)^{10,29–31}, the findings presented here now extend this concept, identifying anisotropic electrostatic interactions as a new, powerful, and externally tunable source of such torques

(unlike vdW and dipole–dipole torques, which originate from the intrinsic properties of the materials). Moreover, mechanism of using competitive ion adsorption to generate tunable electrostatic torques is not limited to the Pt system but represents a generalizable strategy for rationally designing complex nanomaterials, particularly when pH is recognized as another means for manipulating surface potentials through H⁺ and OH⁻ adsorption, because this strategy relies on two ubiquitous features of crystalline materials: the existence of distinct facets and the facet-dependent nature of their surface chemistry. Thus, this approach can be readily extended to other noble metals, oxides, and semiconductors, where ion- or pH-sensitive charge of surface terminations provides a natural handle for electrostatic manipulation^{19,34–36}. While this approach may seem to be inapplicable to cubic systems or any system that is characterized by growth habits for which the faces are crystallographically identical, that is not the case, because all crystallographic directions are represented nonetheless, even if only at rounded corners. Recent results on growth of hematite (Fe₂O₃) by OA show that crystals with very different growth habits all attach along the <001> direction even when there is no (001) facet present and thus attachment must occur on the corners of the nanocrystals⁹

The collective findings in this study demonstrate that the interplay of ion adsorption and resultant surface potential can significantly influence attachment behavior during OA. Anisotropic electrostatic interactions, created by competing ions with facet-dependence, induce approaching NPs to preferentially orient before attachment. This surface potential evolves dynamically in response to changes in the relative surface coverages of the competing ions as surface area decreases during repeated OA events, potentially switching the direction of OA. Unlike intrinsic vdW and dipole–dipole interactions, electrostatic interactions can be finely and dynamically tuned across all surfaces by adjusting solution chemistry, such as pH, electrolyte type, and concentration. While electrostatic interactions have long been recognized as important in NP assembly, their role has typically been considered in terms of static attractive or repulsive forces rather than as dynamic, facet-specific torques capable of switching attachment pathways in real time. This suggests that the dynamic manipulation of these facet-specific electrostatic interactions and resultant torques provides an underexplored strategy for engineering NP assemblies, enabling precise spatial and temporal control over the OA process.”

9. Wang, Y. *et al.* Particle-based hematite crystallization is invariant to initial particle morphology. *Proceedings of the National Academy of Sciences* **119**, e2112679119 (2022).
29. Liu, L. *et al.* Effect of Solvent Composition on Non-DLVO Forces and Oriented Attachment of Zinc Oxide Nanoparticles. *ACS Nano* **18**, 16743–16751 (2024).
30. Zhang, X. *et al.* Direction-specific van der Waals attraction between rutile TiO₂ nanocrystals. *Science* **356**, 434–437 (2017).
34. Su, S. *et al.* Facet-Dependent Surface Charge and Hydration of Semiconducting Nanoparticles at Variable pH. *Advanced Materials* **33**, 2106229 (2021).

35. Liang, Y. *et al.* Facet-dependent surface charge and Pb²⁺ adsorption characteristics of hematite nanoparticles: CD-MUSIC-eSGC modeling. *Environmental Research* **196**, 110383 (2021).

36. Chatman, S., Zarzycki, P. & Rosso, K. M. Surface potentials of (001), (012), (113) hematite (α -Fe₂O₃) crystal faces in aqueous solution. *Phys. Chem. Chem. Phys.* **15**, 13911 (2013).

Minor comments:

Comment 5: In Fig. 4d and e, it would be helpful to draw the reader's attention to the change in axis or keep the axis the same in both so that the different trend is clearly visible.

Response 5: We appreciate the reviewer's feedback. As suggested, we modified (renumbered) Fig. 5d,e to improve clarity by changing the axis label for the fraction using a horizontal line instead of a slash. We also revised the internal annotation of “{100}–{100} alignment” to “ $P_{100} \gg P_{111}$ ” to better reflect the trend and help readers interpret the data more intuitively.

Revised Fig. 5: Evolution of anisotropic surface potential due to repeated OA and transition of pre-aligning NP surface.

a,b, pH (a) and UV-vis absorbance at 236 nm (b) during the growth of Pt mesocrystals at initial concentrations of K_2PtCl_4 and HCOOH of 1.45 mM and 52 mM, respectively. **c,** Zeta potential (ζ) of Pt{100} versus Pt{111} as a function of A_{tot} in solution containing 2.9 mM HCl, 2.9 mM KCl, and 26.5 mM HCOOH; $[H^+]$, $[K^+]$, $[Cl^-]$ and $[HCOO^-]$ are 4.0 mM, 2.9 mM, 5.8 mM and 1.14 mM, respectively. Measurements were taken three times at each point. **d,e,** Alignment ratio as a function of separation distance between two NPs. The ratio of probabilities for the {100}–{100} and {111}–{111} alignments (P_{100}/P_{111}) when $|\zeta_{100}| < |\zeta_{111}|$ (at $A_{tot}=30 m^2/L$ in c) (d) and the ratio

of the probabilities for the $\{111\}$ - $\{111\}$ and $\{100\}$ - $\{100\}$ alignments (P_{111}/P_{100}) when $|\zeta_{100}| > |\zeta_{111}|$ (at $A_{\text{tot}}=3 \text{ m}^2/\text{L}$ in **c**) (**e**).

Comment 6: In Fig. 1e,g circles are drawn around the NPs making up the mesocrystal. I find it difficult to differentiate the individual NP from the image, can the authors annotate the images in a way that the reader can easily identify these?

Response 6: We appreciate the reviewer's feedback. We adjusted the lines indicating the lattice planes and made the dashed circles sparser to improve the visibility of the necks between adjacent NPs at the edge in Fig. 1e,g. We also added a schematic showing the OA direction and the resulting morphology to help understanding.

Revised Fig. 1: Morphological analysis of branched cubic Pt mesocrystals.

a, Ex situ TEM image of synthesized mesocrystals formed from 1.45 mM of K_2PtCl_4 and 52 mM of HCOOH . Inset, schematic of a branched cubic mesocrystals showing the cube-shaped core (yellow) and branches (purple) growing from the cube faces. **b,c**, SAED pattern (**b**) and Scanning TEM image (**c**) from a single mesocrystal. Inset in **b**, corresponding TEM image. Scale bar, 40 nm. **d,e**, HR-TEM images (**d**) and corresponding FFT patterns (**e**) of cubic core obtained at 45 min, with an outline showing $\{100\}$ attachments between NPs. **f,g**, HR-TEM images (**f**) and corresponding FFT patterns (**g**) of branches showing they result from $\{111\}$ attachments between NPs to form $\{111\}$ -aligned nanorods. NPs are indicated by dotted circles and attachment plane is indicated by solid lines in **d,f**. Images in **a–c** are observed 2 days after the reaction. Z.A., zone axis. **h**, Schematics showing the OA direction and the resulting morphology.

Comment 7: Some supplementary figures are not referred to correctly in the main text (example: Supplementary Fig. 15 should be 16). Please check this during revision.

Response 7: We appreciate the reviewer's feedback and apologize for the confusion. After making substantial revisions to the sections "*The role of surface-potential in controlling NP alignment*" and "*The role of ion competition in driving variations in surface potential*", we reviewed the supplementary figures to ensure they are correctly referenced. (Renumbered) Supplementary Figs. 14 and 15 are clearly indicated in the relevant text (Page 5, Line 4; in the main text), as shown below:

“To test this, we measured the zeta potential (ζ)—a proxy for surface potential—of surfaces dominated by {100} and {111} facets as a function of A_{tot} under conditions mimicking the stage ii–iii (Supplementary Figs. 14 and 15). ...”